# MIND: Masked and Inverse Dynamics Modeling for Data-Efficient Deep Reinforcement Learning

## Abstract

In pixel-based deep reinforcement learning (DRL), learning representations of states that change because of an agent's action or interaction with the environment poses a critical challenge in improving data efficiency. Recent data-efficient DRL studies have integrated DRL with self-supervised learning and data augmentation to learn state representations from given interactions. However, some methods have difficulties in explicitly capturing evolving state representations or in selecting data augmentations for appropriate reward signals. Our goal is to explicitly learn the inherent dynamics that change with an agent's intervention and interaction with the environment. We propose masked and inverse dynamics modeling (MIND), which uses masking augmentation and fewer hyperparameters to learn agent-controllable representations in changing states. Our method is comprised of a self-supervised multi-task learning that leverages a transformer architecture, which captures the spatio-temporal information underlying in the highly correlated consecutive frames. MIND uses two tasks to perform self-supervised multi-task learning: masked modeling and inverse dynamics modeling. Masked modeling learns the static visual representation required for control in the state, and inverse dynamics modeling learns the rapidly evolving state representation with agent intervention. By integrating inverse dynamics modeling as a complementary component to masked modeling, our method effectively learns evolving state representations. We evaluate our method by using discrete and continuous control environments with limited interactions. MIND outperforms previous methods across benchmarks and significantly improves data efficiency.

## 1 Introduction

Pixel-based deep reinforcement learning (DRL) has built remarkable agents capable of sequential decision-making using vast amounts of data (Bellemare et al., 2013; Tassa et al., 2018). However, in practice, training an agent with such extensive data is inefficient, time-consuming, and costly. Therefore, it is important for DRL to become more efficient in training agents by using high-dimensional input data with limited interactions within the environment. Recent studies have combined DRL with self-supervised learning (SSL) and data augmentation to improve data efficiency by prompting the learning of effective state representations for given interactions (Laskin et al., 2020a; Yarats et al., 2021b; Schwarzer et al., 2021a; Liu et al., 2021; Zhu et al., 2022).

Existing methods can be categorized as follows: (1) applying suitable data augmentation for the environment without an auxiliary task, (2) defining SSL with contrastive learning as an auxiliary task, and (3) pre-training based on contrastive learning using collected datasets (Laskin et al., 2020a; Yarats et al., 2021b; Schwarzer et al., 2021a; Zhu et al., 2022; Stooke et al., 2021; Liu & Abbeel, 2021; Schwarzer et al., 2021b; Yarats et al., 2021a; Khan et al., 2021). However, these methods demand consideration of numerous factors such as a memory bank, positive or negative samples, and large batch sizes when applying SSL. Data augmentations must be carefully chosen for each environment to avoid providing incorrect reward signals to the agent. Furthermore, because state representations evolve according to an agent's actions or interactions, most methods that focus on visual representation are limited in their ability to explicitly learn drastically changing environments. To this end, we propose a method that reduces the need for other data augmentations and for complex

factors that require tuning. Our approach instead allows for learning representations that an agent can control under evolving states.

In this study, we propose masked and inverse dynamics modeling (MIND) for data-efficient DRL. Our approach is based on two core components: masked modeling and inverse dynamics modeling. Masked modeling has succeeded in fields such as natural language processing and computer vision by learning effective representations of data (Pathak et al., 2016; Devlin et al., 2018; Brown et al., 2020; Conneau & Lample, 2019; He et al., 2022; Bao et al., 2021). Some methods have facilitated learning by pixel-by-pixel reconstruction of masked input data. Masked autoencoders (MAE) (He et al., 2022) learned robust data representations by masking in patch units and reconstructing with a vision transformer (ViT) (Dosovitskiy et al., 2020). These methods were used to pre-train the encoder for downstream tasks. However, their direct application to DRL, which optimizes policies through interaction with the environment, is challenging. Pre-training in DRL requires collection of a dataset. Collecting such datasets with random or expert agents is limited and costly. Recent DRL frameworks are known to use latent state representations that exclude irrelevant information and preserve only essential information for efficient policy learning (Lesort et al., 2018). Therefore, we need a predictive method based on latent representations rather than pixel-based predictions without pre-training.

In DRL, dynamics models have frequently been used as SSL to learn state representations for environments that change with an agent's actions or interactions (Schwarzer et al., 2021a;b; Lesort et al., 2018; Hansen et al., 2020; Seo et al., 2023; Shelhamer et al., 2016). There are two types of dynamics modeling: forward and inverse. Forward dynamics modeling inputs the current state and action into a neural network and predicts the representation of the next state. In contrast, inverse dynamics modeling predicts the current action by entering the current and next states. These methods are easy to implement, and stable learning is achievable without representation collapse becoming a classification and regression problem (Lesort et al., 2018; Schwarzer et al., 2021b; Burda et al., 2019). However, forward dynamics modeling can generate trivial solutions, such as constant zero features, for states when predicting the next state in pixel space (Hansen et al., 2020). In addition, when dynamics modeling is applied at a single point in time, there are limitations in learning representations because of the nature of DRL, which learns through sequential decision-making. We propose inverse dynamics modeling, which is effective for explicitly learning representations of important states by encoding essential information to consecutively predict actions that cause a change in the state while simultaneously discarding irrelevant information (Lesort et al., 2018).

Our method uses self-supervised multi-task learning, which simultaneously trains masked and inverse dynamics modeling. Masked modeling, which focuses on visual elements, has limitations in learning representations in an environment with rapidly changing backgrounds, viewpoints, obstacles, and objects. To cope with these limitations, MIND combines masked and inverse dynamics modeling to learn not only visual representations but also state representations that evolve according to agent intervention. MIND incorporates transformers (Vaswani et al., 2017) that can capture spatio-temporal information for learning highly correlated consecutive frames of the environment. Masked modeling involves reconstructing masked consecutive states into complete consecutive states on latent representations. Because this method predicts consecutive states, actions can also be considered as embedded, thus allowing it to encode agent-controllable state information for learning representations (Lesort et al., 2018). Inverse dynamics modeling predicts consecutive actions by using masked consecutive states and consecutive original next states. It effectively learns by constraining state representations to predict consecutive actions that cause state changes. MIND is a simple and effective method that relies solely on masking augmentation, eliminating the need for complex factors in learning representations. Moreover, MIND achieves computationally effective learning by using low-dimensional latent representations for auxiliary tasks, while considering the high correlation of consecutive frames. Our method is combined with Rainbow (Hessel et al., 2018) and soft actor-critic (SAC) (Haarnoja et al., 2018) among other DRL methods to simultaneously optimize representation learning and policy. MIND can be applied in both discrete and continuous action spaces. We thoroughly evaluate across 26 games on Atari (Łukasz Kaiser et al., 2020) (discrete) and six DeepMind control suite (DMControl) environments (Tassa et al., 2018) (continuous). The interactions were limited to 100k steps. Our method outperforms previous methods in evaluation metrics and demonstrates near-human performance.

## 2 RELATED WORKS

### 2.1 DATA-EFFICIENT REINFORCEMENT LEARNING

Many studies have been proposed to encourage state representation learning of DRL to improve data efficiency. Representatively, EfficientZero, a model-based DRL, significantly reduced data complexity in an end-to-end manner by applying SSL and data augmentation (Ye et al., 2021). CURL (Laskin et al., 2020a) sought to optimize representation learning and policy by combining MoCo (He et al., 2020) with Rainbow during contrastive learning. DrQ (Yarats et al., 2021b) used only data augmentation and a regularized value function without auxiliary loss or pre-training. RAD (Laskin et al., 2020b) enhanced performance by applying data augmentation to the pixel-based states, with empirical reports identifying suitable data augmentation. SODA (Hansen & Wang, 2021) proposed a learning process that separated representation learning and DRL using augmented and non-augmented data to prevent the risk of directly learning policy in severely transformed states. RCRL (Liu et al., 2021) introduced contrastive learning to discriminate between state-action pairs with different rewards. However, these methods have difficulty in considering numerous factors when applying SSL, and data augmentation must be carefully chosen to avoid providing incorrect reward signals to the agent.

### 2.2 MASKED MODELING IN REINFORCEMENT LEARNING

Masked modeling has been used in many studies as a representation learning function to improve the performance of DRL. M-CURL (Zhu et al., 2022) has been proposed as a way to perform contrastive learning by adding a transformer to consider the high correlation between consecutive states and randomly masked input states. MLR (Yu et al., 2022) was combined with DRL as an auxiliary task to predict complete state representations in latent space by applying spatio-temporal cube masking to consecutive states. MWM (Seo et al., 2023) proposed a model-based DRL framework that separated representation learning and dynamics learning. Notably, this method showed the effectiveness of ViT for DRL, which reconstructs visual observations with patch-level masking from feature maps. MTM (Wu et al., 2023), as a pre-training method to reconstruct randomly masked sequence trajectory segments, has been successfully applied to several tasks with various capabilities including future prediction, imitation learning, and representation learning. Inspired by the success of these methods, we adopted masked modeling, which simply and effectively learns visual representations that agents can control to improve data efficiency.

### 2.3 DYNAMICS MODELING IN REINFORCEMENT LEARNING

Dynamics modeling has been widely used as a self-supervision task in DRL to capture dynamic information inherent in the environment. PBL (Guo et al., 2020) proposed multistep prediction of future latent observations and reverse prediction from latent observations to states as representation learning to capture information about environmental dynamics. PlayVirtual (Yu et al., 2021) proposed to predict a latent future state based on the current state and action, and then predicts the previous state using a backward dynamics model. SPR (Schwarzer et al., 2021a) improved performance with a multistep forward dynamics model by adding an action-conditioned transition model. SGI (Schwarzer et al., 2021b) combined SPR, inverse dynamics modeling, and goal-conditioned RL to improve data efficiency by pre-training on reward-free data and subsequent fine-tuning in task-specific environments. PAD (Hansen et al., 2020) proposed a framework that can be learned in a new environment even after deploying a policy learned by SSL leveraging dynamics modeling. SMART (Sun et al., 2022) introduced a reward-agnostic control transformer, which was pre-trained with dynamics modeling and showed superior performance in downstream tasks. In model-based DRL frameworks, several methods that proposed task-conditioned inverse dynamics modeling in latent space (Paster et al., 2020), dynamics modeling with controllable and uncontrollable information in isolated states (Pan et al., 2022), and the autoregressive transformer that capture dynamics (Micheli et al., 2022) have successfully built world models. More recently, AC-State (Lamb et al., 2022) and ACRO (Islam et al., 2022) proposed multi-step inverse dynamics that can convey only the information needed while excluding information that is not relevant to agent control. To enhance data efficiency, we adopted inverse dynamics modeling, which learns state representations by capturing sufficient information about changing factors in the environment. We constructed a method

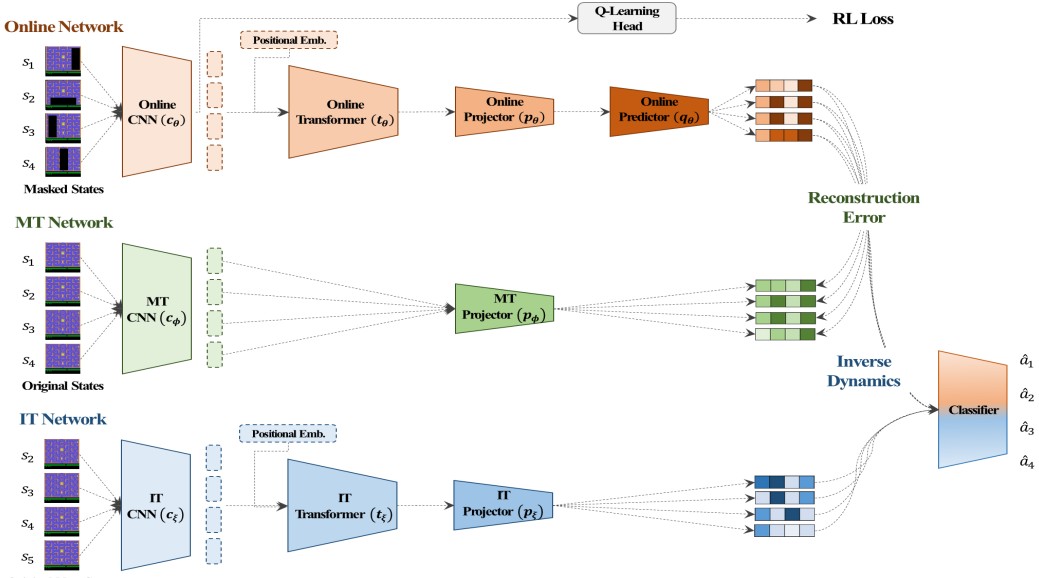

Figure 1: An overall architecture of MIND. MIND performs masked modeling and inverse dynamics modeling simultaneously. The masked consecutive states are fed into the online network. Masked modeling reconstructs the consecutive original states by inputting them into the MT network. Inverse dynamics modeling inputs a series of original next states into the IT network and predicts consecutive actions with a classifier.

for predicting consecutive actions for consecutive states with a consideration of the high correlation between consecutive frames.

## 3 PROPOSED METHOD

In this section, we describe MIND in detail. Figure 1 illustrates its overall architecture. Our goal is to elevate data efficiency by learning high-quality state representations with transitions obtained from the environment despite limited interactions. MIND benefits from multi-task learning, utilizing the relationships between tasks to boost performance in each individual task. Masked modeling predicts masked consecutive states with implicitly embedded actions, enabling the agent to learn controllable state representations. However, its focus on visual representations limits its ability to capture information about environmental changes. We combine masked modeling with inverse dynamics modeling to improve data efficiency by learning crucial information about the visual and changing factors of the state through self-supervised multi-task learning. In addition to this, we include the transformer that can capture spatio- temporal information to understand the high correlation of consecutive frame. MIND uses masking augmentation exclusively for consecutive states as input into the online network without the need to consider other augmentation.

We use masked modeling as an auxiliary task to reconstruct masked consecutive states for state representation learning. Conventional masked modeling predicts raw pixels from random pixel levels of masked images. However, pixel-level prediction tends to base its prediction on the averages of the pixel distribution, limiting its predictive accuracy. To avoid this problem, we force the reconstruction to be summarized low-dimensional latent representations rather than pixel-level reconstructions (Lesort et al., 2018). Our method consists of an online network and a masked target (MT) network in Figure 1. Each network is parameterized with $\theta$ and $\phi$. Both networks have the same convolutional neural network (CNN) and projector; only the online network includes a transformer and predictor. Masked modeling uses consecutive original state representations of the MT network as the target values to be predicted from masked consecutive state representations output from the online network. The target values do not require a transformer that considers spatio-temporal information as labels for the missing state representations. Because our method's reconstructions use the output of consecutive representations from each network, it can be seen that implicitly embedded actions facilitate

state representation learning by encoding agent-controllable state information (Lesort et al., 2018). The parameter $\theta$ is trained with the following mean squared error (MSE) loss, and $\phi$ is updated with the exponential moving average (EMA) of the parameter $\theta$ with the momentum coefficient $\beta$:

$$\mathcal{L}_{recon} = \frac{1}{T} \sum_{i=1}^{T} (p_\phi(c_\phi(s_i)) - q_\theta(p_\theta(t_\theta(c_\theta(\mathcal{M} \odot s_i)))))^2, \tag{1}$$

$$\phi \leftarrow \beta\phi + (1-\beta)\theta, \tag{2}$$

where $T$ is the maximum length of the input sequence. The CNN, transformer, projector, and predictor are denoted by $c$, $t$, $p$, and $q$, respectively. $\mathcal{M}$ is a binary mask corresponding to the state domain and $\odot$ is the element-wise multiplication. $\beta$ is a value between zero and one.

In the proposed method, we use inverse dynamics modeling, the concept of predicting the current action $a_t$ from the current state $s_t$ and the next state $s_{t+1}$, as a self-supervised task. Inverse dynamics modeling explicitly learns effective state representations by encoding essential information for the task of predicting action that produces the information on changes in state. Any irrelevant information is discarded (Lesort et al., 2018). We construct a new architecture that predicts consecutive actions from low-dimensional latent representations, learning state representations by capturing the inherent information that changes in response to agent intervention. The proposed method consists of online and inverse target (IT) networks parameterized by $\theta$ and $\xi$, as shown in Figure 1. Both networks have the same CNN, transformer, and projector, with a predictor added only to the online network. To predict the consecutive actions, the online network uses masked consecutive states, and the IT network uses the next consecutive original states. We encode relative temporal positional information into each feature vector by adding a positional embedding to consider the relative position of the output feature vectors from the CNN before inputting them to the transformer. Our method promotes learning by leveraging state representations to predict consecutive actions using a classifier based on the consecutive representations generated by each network. In particular, our method effectively captures past or future state dependencies by integrating information across input sequence states through the attention mechanism of the transformer. The parameter $\theta$ is trained with cross-entropy loss and $\xi$ is updated with the EMA of the parameter $\theta$, using the momentum coefficient $\beta$:

$$\mathcal{L}_{inverse} = -\frac{1}{T} \sum_{j=1}^{T} \sum_{i=1}^{k} a_{ji} \log(P_{ji}), \tag{3}$$

$$\xi \leftarrow \beta\xi + (1-\beta)\theta, \tag{4}$$

where $T$ is the maximum length of the input sequence and $k$ is the number of classes. $a_{ji}$ is the true label for the action on the $j$th data and $P_{ji}$ is the softmax probability for the $i$th class on the $j$th data. $\beta$ is a value between zero and one.

Our method is based on a self-distillation approach consisting of online and target networks. This method benefits from asymmetric architecture and EMA updates (Balestriero et al., 2023). The predictor serves a critical role as a whitening operation, alleviating the problem of representation collapse because of the distributing effect over batch samples (Ermolov et al., 2021). Moreover, the efficiency of the self-distillation method is evident in its ability to prevent mode collapse by updating one network's weights using an EMA of the other's. Furthermore, our method encourages computationally efficient policy and representation learning by using low-dimensional latent representations when performing auxiliary tasks while taking into consideration the high correlation of consecutive frames. The DRL task uses the output feature vector in the Q-learning head by inputting a single state to the online CNN that operates concurrently with MIND. Therefore, MIND, as an auxiliary task that is optimized along with policy learning, is learned with the following total loss for agent training:

$$\mathcal{L}_{total} = \mathcal{L}_{rl} + \lambda(\mathcal{L}_{recon} + \mathcal{L}_{inverse}), \tag{5}$$

where $\mathcal{L}_{rl}$ is the loss function of the Rainbow (Hessel et al., 2018) or SAC (Haarnoja et al., 2018). $\lambda$ is a balancing hyperparameter.

## 4 EXPERIMENTS

**Environment and evaluation**   We evaluate the data efficiency of MIND on the 26 discrete control benchmark Atari games and the six continuous control benchmark DMControl. These benchmarks

Table 1: Performance results of the human scores, previous methods, and MIND for 26 games on Atari 100k. We reported the mean and median HNS using the reported scores from previous methods and human scores, which is an official measure. We also reported the standard deviation for HNS, the number of games won over humans (# Super Human), and the number of games won between methods (# Games Won).

| Game | Random | Human | DER | OTR | CURL | DrQ | SPR | PlayVirtual | MLR | MIND |
|---|---|---|---|---|---|---|---|---|---|---|
| Mean HNS | 0.000 | 1.000 | 0.350 | 0.217 | 0.261 | 0.369 | 0.616 | 0.637 | 0.679 | **0.735** |
| Median HNS | 0.000 | 1.000 | 0.189 | 0.128 | 0.092 | 0.212 | 0.396 | **0.472** | 0.428 | 0.416 |
| Standard Deviation | 0 | 0 | ± 0.436 | ± 0.507 | ± 0.558 | ± 0.470 | ± 0.805 | ± 0.738 | ± 0.756 | ± 0.740 |
| # Super Human | 0 | N/A | 2 | 1 | 2 | 2 | 6 | 4 | 6 | **9** |
| # Games Won | N/A | N/A | 0 | 0 | 0 | 2 | 3 | 3 | 6 | **13** |

are suitable for evaluating data efficiency, as each environment consists of diverse observations and tasks, allowing only 100k steps of interaction (Laskin et al., 2020a; Schwarzer et al., 2021b). On Atari, we calculate and report the mean and median human-normalized score (HNS), which are used to measure the performance on each game. The HNS is calculated as $\frac{(agent\,score - random\,score)}{(human\,score - random\,score)}$ and aggregated over the 26 games by mean or median. Furthermore, we use robust evaluation metrics that address bias issues without being unduly influenced by outliers in high variability benchmarks such as Atari (Agarwal et al., 2021). These evaluation metrics include interquartile mean (IQM) and optimality gap (OG), both calculated using stratified bootstrap confidence intervals. The IQM is computed by discarding the top and bottom 25% of the runs and averaging the remaining 50% of the runs. The OG quantifies the extent to which an agent falls short of the minimum default score for human-level performance. Higher IQM and lower OG values are preferable. These are calculated based on the results of multiple runs, with models trained on different seeds. Each run is evaluated using the average score over 100 episodes. On DMControl, we evaluate the performance of the agent with the mean and median scores over ten episodes. The score of each environment ranges from zero to 1,000.

**Implementation**   We combine MIND with the DRL method for end-to-end optimization. The DRL methods, Rainbow (Hessel et al., 2018) and SAC (Haarnoja et al., 2018), are used for discrete control and continuous control benchmarks, respectively. Detailed information on Rainbow, SAC, and modelings of MIND is available in Appendices A.2, A.3, A.4, and A.5. Unlike previous methods (Yarats et al., 2021b; Schwarzer et al., 2021a; Laskin et al., 2020b; Yu et al., 2021), MIND applies only masking augmentation to the input state of the online network. This masking strategy removes square-shaped information through uniform sampling within a range of hyperparameter scale and ratio values at each training iteration. We fix the minimum value of the scale and ratio to the default value, corresponding to about 2%, and only change the maximum value. The maximum value uses a scale of 0.33 and a ratio of 3.3, corresponding to 20%. In addition, we set the main hyperparameters of MIND, the input sequence length, and transformer depth, to six and two, respectively. The momentum coefficient was 0.999 in Atari and 0.95 in DMControl. As demonstrated in Equation 5, we balance the loss functions of DRL methods and MIND using a weighted $\lambda$, where $\lambda = 1$ has proven effective in most environments. Further implementation details can be found in Appendix B.

**HNS Atari results**   We compared MIND with model-free data-efficient methods such as DER (Van Hasselt et al., 2019), OTR (Kielak, 2019), CURL (Laskin et al., 2020a), DrQ (Yarats et al., 2021b), SPR (Schwarzer et al., 2021a), PlayVirtual (Yu et al., 2021), and MLR (Yu et al., 2022). All of these methods are built upon Rainbow, with average scores reported from DER, OTR, CURL, DrQ, and SPR based on 100 random seeds. PlayVirtual and MLR reported their results based on 15 and three random seeds, respectively, while MIND reported results based on ten random seeds. Table 1 presents the aggregated results of MIND and previous methods for the 26 games on the Atari 100k. Across all methods, MIND was clearly superior in terms of the highest score per game, outperforming humans in nine games. When compared with the mean and median HNS, our method surpassed the performance of previous methods. The median HNS was lower than PlayVirtual and MLR. However, because the mean and median results are influenced by games with either very high HNS scores, such as Crazy Climber, Kangaroo, and Krull, or very low HNS scores, such as Asterix, Chopper Command, Private Eye, and Seaquest, it is crucial to also evaluate these games with objective metrics.

**IQM and OG Atari results** To make meaningful comparisons, we used robust evaluation metrics that address bias issues without being unduly influenced by outliers (Agarwal et al., 2021). Figure 2 shows the results of MIND and previous methods in terms of IQM and OG, both calculated using stratified bootstrap confidence intervals. MIND achieved an IQM of 0.500, which is 15.7% higher than MLR of 0.432, 33.3% higher than SPR of 0.375, and 29.9% higher than PlayVirtual of 0.385. In terms of OG, MIND showed superior performance with 0.489, lower than MLR of 0.522, SPR of 0.554, and PlayVirtual of 0.552. Therefore, our method demonstrated the highest data efficiency and was the closest to human-level

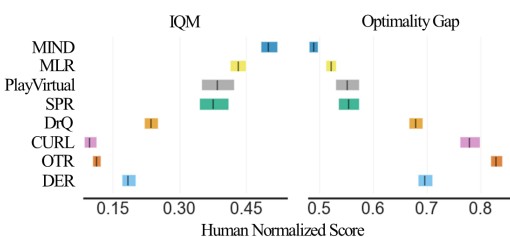

Figure 2: Comparison results between methods in the Atari 100k with IQM and OG by using stratified bootstrap confidence levels. The full scores of all methods can be found in Appendix C.1.

performance. Moreover, our method performed robustly in environments with rapidly changing backgrounds, viewpoints, obstacles, and objects, such as Alien, Crazy Climber, Freeway, and Kung Fu Master.

**Atari performance profiles** We also presented performance profiles (Dolan & Moré, 2002) using HNS with 95% confidence intervals. Figure 3 illustrates the performance profiles of the Atari 100k. The shaded regions represent 95% confidence interval bands based on stratified bootstrapping. These results depict a tailed distribution of scores across tasks, where higher curves represent better performance (Agarwal et al., 2021). Performance profiles for the distribution X are computed as $\hat{F}_X(\tau) = \frac{1}{M} \sum_{m=1}^{M} \frac{1}{N} \sum_{n=1}^{N} \mathbb{1}[x_{m,n} > \tau]$ and represent the fraction of runs above the score $\tau$ over $N$ runs and $M$ tasks. The intersection point at the value of 0.5 represents the median. Through these results, we demonstrated that MIND outperforms previous methods in terms of performance profiles.

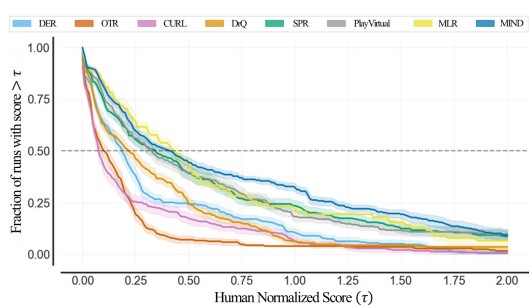

Figure 3: Performance profiles of methods based on HNS tail distributions on Atari 100k.

**DMControl-100k results** We compare all methods for six environments on DMControl in Table 2. Comparative methods include PlaNet (Hafner et al., 2019b), Dreamer (Hafner et al., 2019a), SAC+AE (Yarats et al., 2021c), SLAC (Lee et al., 2020), CURL (Laskin et al., 2020a), RAD (Laskin et al., 2020b), DrQ (Yarats et al., 2021b), PlayVirtual (Yu et al., 2021), and MLR (Yu et al., 2022). All results reported average scores over ten runs using different random seeds. Our method outperformed previous methods on mean and median scores for the entire environment. In particular, MIND was slightly higher than the median of the previous best method, MLR. This shows that the variance in scores for each environment is relatively small. Therefore, we demonstrated the high

Table 2: Performance comparison of DMControl 100k. The best score of each environment is in **bold**.

| Environment | PlaNet | Dreamer | SAC+AE | SLAC | CURL | RAD | DrQ | PlayVirtual | MLR | MIND |
|---|---|---|---|---|---|---|---|---|---|---|
| Finger, spin | $136 \pm 216$ | $341 \pm 70$ | $740 \pm 64$ | $693 \pm 141$ | $767 \pm 56$ | $856 \pm 73$ | $901 \pm 104$ | $\mathbf{915 \pm 49}$ | $907 \pm 58$ | $863 \pm 68$ |
| Cartpole, swingup | $297 \pm 39$ | $326 \pm 27$ | $311 \pm 11$ | - | $582 \pm 146$ | $\mathbf{828 \pm 27}$ | $759 \pm 92$ | $816 \pm 36$ | $806 \pm 48$ | $823 \pm 35$ |
| Reacher, easy | $20 \pm 50$ | $314 \pm 155$ | $274 \pm 14$ | - | $538 \pm 233$ | $826 \pm 219$ | $601 \pm 213$ | $785 \pm 142$ | $866 \pm 103$ | $\mathbf{929 \pm 46}$ |
| Cheetah, run | $138 \pm 88$ | $235 \pm 137$ | $267 \pm 24$ | $319 \pm 56$ | $299 \pm 48$ | $447 \pm 88$ | $344 \pm 67$ | $474 \pm 50$ | $482 \pm 38$ | $\mathbf{513 \pm 31}$ |
| Walker, walk | $224 \pm 48$ | $277 \pm 12$ | $394 \pm 22$ | $361 \pm 73$ | $403 \pm 24$ | $504 \pm 191$ | $612 \pm 164$ | $460 \pm 173$ | $\mathbf{643 \pm 114}$ | $628 \pm 127$ |
| Ball in cup, catch | $0 \pm 0$ | $246 \pm 174$ | $391 \pm 82$ | $512 \pm 110$ | $769 \pm 43$ | $840 \pm 179$ | $913 \pm 53$ | $926 \pm 31$ | $933 \pm 16$ | $\mathbf{950 \pm 21}$ |
| Mean | 135.8 | 289.8 | 396.2 | 471.3 | 559.7 | 716.8 | 688.3 | 729.3 | 772.8 | **784.4** |
| Median | 137.0 | 295.5 | 351.0 | 436.5 | 560.0 | 827.0 | 685.5 | 800.5 | 836.0 | **842.9** |

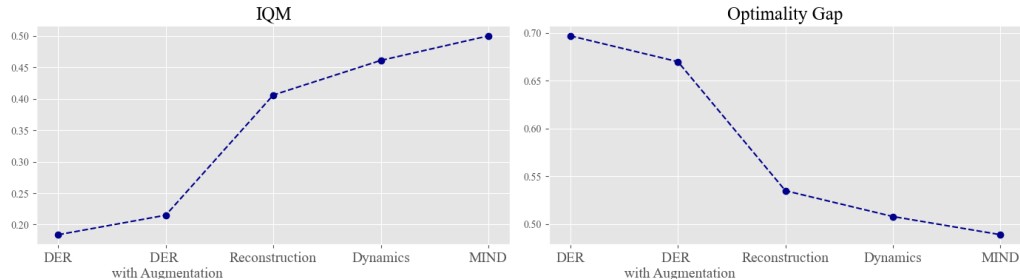

Figure 4: Comparison results between IQM and OG to determine the impact of applying augmentation, masked modeling, inverse dynamics modeling, and MIND to the DRL method, respectively.

data efficiency of MIND because it outperformed previous methods in Reacher, easy; Cheetah, run, and Ball in cup, catch. Furthermore, we extended our experiments to the medium-difficulty DM-Control tasks and compared MIND with state-of-the-art methods. Detailed results can be found in Appendix C.2.

## 5 ANALYSIS

In this section, we demonstrate the effectiveness of modelings of MIND, components for architecture design, and masking strategies. We performed further analyses on the Atari 100k and computed five runs based on five random seeds. Each run was evaluated with 100 episodes. MIND used the main results. The default values of key hyperparameters used in experiments, such as input sequence length, transformer depth, maximum masking ratio, and momentum coefficient, were set to six, two, 20%, and 0.999.

**Effectiveness of combined modeling and augmentation** We hypothesized that adopting only masking augmentation on the input state and combining inverse dynamics modeling and masked modeling to learn evolving state factors would yield better representations than applying these tasks individually. Figure 4 shows the IQM and OG results obtained from multiple tests by combining each task with Rainbow. It clearly indicates that the augmentation in the DRL method is effective in its own right. However, we observed that the aspect of reconstruction contributed more to performance improvement than augmentation alone. Despite its merits, masked modeling encountered difficulties in learning representations of evolving states because of its focus on static visual representations. In addition, the application of inverse dynamics modeling alone appeared to reflect information about the rapidly changing environment to a certain extent. However, predictive accuracy remains uncertain as the input consists of masked consecutive states with missing information, such as the agent and obstacles. Therefore, our results affirmed our initial hypothesis that the benefits of multi-task learning, combined with the ability to use the relationship between masked modeling and inverse dynamics, can improve the data efficiency of DRL.

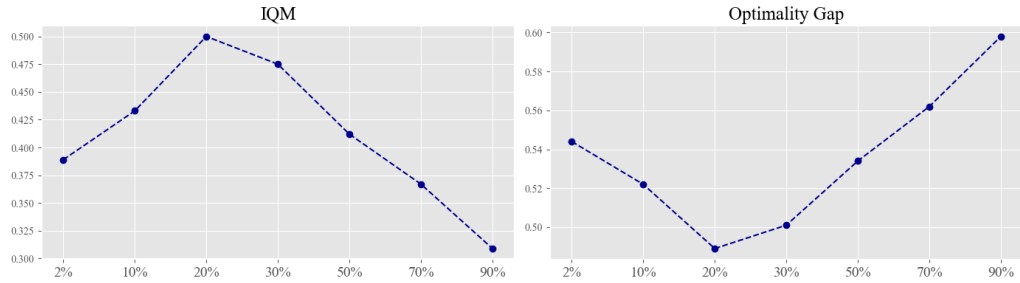

Figure 5: Comparison results of IQM and OG by applying various maximum masking ratios to consecutive states.

**Maximum masking ratio**   The masking ratio is a crucial factor in determining performance. It is essential to identify the appropriate ratio for state representation learning, as an excessively high masking ratio can obscure important object information such as the agent, making goal achievement challenging. We investigated the effect of different maximum masking ratios (2%, 10%, 20%, 30%, 50%, 70%, and 90%). Figure 5 shows the aggregated results of multiple tests for variations in the maximum masking ratio. Ratios of 2% and 10% indicate that the representation learning task is hindered because it is easier to reach the goal by capturing detailed visual signals. Notably, using high masking ratios of 50%, 70%, and 90% reveals that most of the important information is obscured, thus reducing the efficiency of the reconstruction task and making it difficult to learn representations of evolving states. Therefore, we confirmed that our method can enhance the data efficiency of DRL by learning representation while preserving important information when the maximum ratio is 20%.

**Masking strategy**   There are various types of masking augmentations available. We compared the random erasing that we used with spatial masking, which independently applies random masks patches for each frame, as presented by MLR. Additionally, we implemented random cropping alongside random erasing to determine its impact. We found that using only random erasing without random cropping was superior. Detailed experimental results and analyzes are written in Appendix C.3.

**Momentum coefficient**   Our method uses target networks and updates them with EMA. The process is quite sensitive to the selection of momentum coefficient (Grill et al., 2020). To assess this, we tested the performance with various momentum coefficients (0.9, 0.99, and 0.999). In conclusion, our method works most efficiently with a coefficient of 0.999 (see Appendix C.3).

**Impact of the target network**   Our MIND takes advantage of a self-distillation approach that involves both online and target networks. This strategy gains from having an asymmetric architecture and updates through EMA (Balestriero et al., 2023). Given that we are already aware of the performance of MLR, we conducted an experiment using only inverse dynamics modeling, both with and without a target network. We found that inverse dynamics modeling without a target network can cause representation collapse. Experimental results and analyses are in Appendix C.3.

**Sequence length and transformer depth**   The sequence length and transformer depth are important hyperparameters of our method. Our tests revealed that the performance improved when we set the length to six and the depth to two. More detailed results and analyses are described in Appendix C.3.

**Discussions**   We provide further discussion about MIND, with more detailed analysis available in Appendix C.4. Because both masked modeling and inverse dynamics modeling can be regarded as regression and classification tasks, we verified the quality of the learned representations using regression and classification evaluation metrics. In addition, we explored the relationship between MLR and MIND. Although both methods are excellent at improving data efficiency, there are differences in the masking strategy and the use of actions embedded in feature vectors. Finally, we delve into the applications and limitations of MIND.

## 6   CONCLUSIONS

In this study, we proposed MIND, a state representation learning approach designed to improve the data efficiency of DRL. MIND operates as the self-supervised multi-task learning that simultaneously learns masked and inverse dynamics modeling tasks. By focusing on visual representation, our method captures the essential information of changing factors by integrating inverse dynamics modeling as a complementary advantage to masked modeling, which struggles to encode information about environmental changes. MIND solely uses masking augmentation and does not require consideration of complex factors in learning representations. Moreover, by considering the high correlation of consecutive frames, MIND achieves efficiency in computational and policy learning through low-dimensional latent representations when performing auxiliary tasks. We demonstrated the superiority of MIND through comprehensive experiments on the Atari 100k and DMControl 100k. MIND showed significant improvements over previous methods in several evaluation metrics. We also conducted ablation studies crucial for designing MIND and provided suitable indicators.

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

## A PRELIMINARIES

### A.1 MARKOV DECISION PROCESS

DRL aims to learn an agent's policy through sequential decision-making by interacting with the environment. DRL can be formalized as a Markov decision process (MDP) defined by the tuple $T = (S, A, P, R, \gamma)$. Here, $S$ represents a set of states, $A$ denotes the set of actions, and $P : S \times A \times S \to [0, 1]$ indicates the transition probability distribution. $R : S \times A \to \mathbb{R}$ stands for the reward function and $\gamma$ is the discount factor. At time step $t$, as a function formulated by $P(s_{t+1}|s_t, a_t)$ according to the Markov property, the agent takes an action $a_t \in A$ from a state $s_t \in S$ and obtains the next state $s_{t+1} \in S$. Reward $R_t = R(s_t, a_t)$ is determined based on the state $s_t$ and action $a_t$. The ultimate goal of the DRL is to find a policy $\pi$ to maximize the expectation of the defined discounted return $G_t = \sum_{k=0}^{T} \gamma^{(k)} R_{t+k+1}$ in a finite MDP environment (Mnih et al., 2013; Sutton & Barto, 2018).

### A.2 DEEP Q-LEARNING

Deep Q network (DQN) (Mnih et al., 2015) is a representative method for solving the discrete action space benchmark, which approximates the Q-function by combining off-policy Q-learning and neural networks. The neural network $Q_\theta$, parameterized by $\theta$, represents the action-value function $Q_\theta^\pi(s, a) = E_{(s,a,s') \sim \mathcal{D}} \big[ R(s, a) + \gamma E_{a' \sim \pi}[Q_\theta^\pi(s', a')] \big]$ over policy $\pi$. Policy $\pi$, which can be inferred from the Q-value, is trained by minimizing the following loss between the predicted value from $Q_\theta$ and the target value estimated by the previous version of the network $Q_\delta$:

$$\mathcal{L}_\theta = \mathbb{E}_{(s,a,s') \sim \mathcal{D}}[(Q_\theta(s, a) - (R(s, a) + \gamma \max_{a'} Q_\delta(s', a'))^2], \tag{6}$$

where $\mathcal{D}$ denotes the replay buffer. $\delta$ is a fixed parameter when network $Q_\theta$ updates multiple iterations. The Rainbow (Hessel et al., 2018) that we used, was developed to improve data efficiency while incorporating enhancements from DQN. The Rainbow applies to the discrete action benchmark Atari.

### A.3 SOFT ACTOR-CRITIC

SAC (Haarnoja et al., 2018) is an off-policy actor-critic method based on the maximum entropy DRL framework (Ziebart et al., 2008). It maximizes the weighted objective of reward and policy entropy to encourage exploration and is robust to noise. SAC updates parameters with soft policy evaluation and soft policy improvement. At the soft policy evaluation step, the soft Q-function with parameter $\theta$ is updated by minimizing the following soft Bellman error:

$$\mathcal{L}_{critic} = \mathbb{E}_{(s,a,s',r) \sim \mathcal{D}}[(Q_\theta(s, a) - R(s, a) - \gamma \bar{V}(s))^2], \tag{7}$$

where $\bar{V}(s) = \mathbb{E}_{a \ \pi} \big[ Q_{\bar{\theta}}(s, a) - \alpha \log \pi(a|s) \big]$ and $\mathcal{D}$ is the replay buffer. $\bar{\theta}$ is the target Q-function parameters and $\alpha$ is the temperature parameter. At the soft policy improvement step, the policy $\pi$ is updated by minimizing the following objective:

$$\mathcal{L}_{actor} = \mathbb{E}_{s \sim \mathcal{D}, a \sim \pi}[(\alpha \log \pi(a|s) - Q_\theta(s, a)]. \tag{8}$$

SAC is used as our backbone model for the continuous action benchmark DMControl.

### A.4 MASKED MODELING

Masked modeling is an SSL method that trains reconstructions from masked images to original images. In general, it involves taking a random pixel-level masked image as input data and reconstructing the raw pixels. Networks typically have an encoder that summarizes the input data and a decoder that restores it to its original size. For instance, $\mathcal{M}$ is a binary mask corresponding to the image area, and $x$ refers to the image. Given a network $f$, the method is trained based on the following mean squared error loss:

$$\mathcal{L}_{recon}(x) = ||\mathcal{M} \odot (x - f((1 - \mathcal{M}) \odot x))||_2, \tag{9}$$

where $\odot$ is the element-wise multiplication.

---

**Algorithm 1** MIND

---

1: Denote CNN $c$, transformer $t$, projector $p$ and predictor $q$
2: Denote parameters of online network as $\theta$, MT network as $\phi$, IT network as $\xi$
3: Denote parameters of Q-learning head as $\psi$
4: Denote maximum time length as $T$, binary mask as $\mathcal{M}$, batch size as $N$
5: Initialize replay buffer $D$
6: **while** Training **do**
7:     Collect experience $(s, a, r, s')$ with $(\theta, \psi)$ and add to replay buffer $D$
8:     Sample a minibatch of sequences of $(s_{1:T}, a_{1:T}, r_T, s'_{1:T}) \sim D$
9:     Masked states $ms_{1:T} = \mathcal{M} \odot s_{1:T}$                       ▷ element-wise multiplication
10:     **for** $\omega$ iterations **do**
11:         $\mathcal{L}^{\omega} \leftarrow 0$
12:         **Calculate** $z_{1:T}^{\omega,o} \leftarrow q_{\theta}(p_{\theta}(t_{\theta}(c_{\theta}(ms_{1:T}))))$     ▷ representations via online network
13:         **Calculate** $z_{1:T}^{\omega,MT} \leftarrow p_{\phi}(c_{\phi}(s_{1:T}))$         ▷ representations via MT network
14:         **Calculate** $z_{1:T}^{\omega,IT} \leftarrow p_{\xi}(t_{\xi}(c_{\xi}(s'_{1:T})))$        ▷ representations via IT network
15:         **Calculate** $\mathcal{L}_{recon}^{\omega} \leftarrow \frac{1}{T}\sum_{j=1}^{T}(z_j^{\omega,MT} - z_j^{\omega,o})^2$        ▷ masked modeling loss
16:         **Calculate** $\mathcal{L}_{inverse}^{\omega} \leftarrow -\frac{1}{T}\sum_{j=1}^{T}\sum_{i=1}^{k} a_{ji}\log(P(\hat{a}_{ji}|z_j^{\omega,o}, z_j^{\omega,IT}))$        ▷ inverse dynamics loss
17:         $\mathcal{L}^{\omega} \leftarrow \lambda(\mathcal{L}_{recon}^{\omega} + \mathcal{L}_{inverse}^{\omega}) + \text{RL loss}(s_T, a_T, r_T, s'_T; \theta)$
18:     **end for**
19:     $\mathcal{L}_{total} \leftarrow \frac{1}{N}\sum_{\omega=0}^{N}\mathcal{L}^{\omega}$
20:     Update $\theta, \psi \leftarrow \text{optimize}((\theta, \psi), \mathcal{L}_{total})$     ▷ update online parameters and Q-learning head
21:     $\phi \leftarrow \beta\phi + (1-\beta)\theta, \xi \leftarrow \beta\xi + (1-\beta)\theta$     ▷ update MT and IT parameters
22: **end while**

---

### A.5 INVERSE DYNAMIC MODELING

Inverse dynamics modeling is a representative method for learning the inherent dynamics in the environment with powerful state representation learning for DRL. Given the transition $\{s_t, a_t, s_{t+1}\}$ in timestep $t$, inverse dynamics modeling predicts the current action $\hat{a}_t$ by inputting current state $s_t$ and next state $s_{t+1}$ into the $f_{\theta}$. This method computes the network $f_{\theta}$ by the following equation:

$$\mathcal{L}_{inverse} = -\sum_{i=1}^{k} t_i \log(P_i), \tag{10}$$

where $t_i$ is the true label for the action and $P_i$ is the softmax probability for the $i$th class.

## B IMPLEMENTATION DETAILS

The MIND implementation is based on open-source CURL[1] (Laskin et al., 2020a), MLR[2] (Yu et al., 2022) and PyTorch (Paszke et al., 2019). Our method evaluates the Atari (Łukasz Kaiser et al., 2020) and DMControl benchmarks primarily used in DRL. Statistical evaluation approach uses open-source rliable[3] (Agarwal et al., 2021).

### B.1 NETWORK ARCHITECTURES

Our self-supervised multi-task learning method, which incorporates masked modeling and inverse dynamics modeling, consists of online, MT, and IT networks. The online network uses a CNN $c$, transformer $t$, projector $p$, and predictor $q$, all parameterized by $\theta$. The MT network utilizes a CNN

---

[1]Link: `https://github.com/aravindsrinivas/curl_rainbow`, licensed under the MIT License.
[2]Link: `https://github.com/microsoft/Mask-based-Latent-Reconstruction`, licensed under the MIT License.
[3]Link: `https://github.com/google-research/rliable`, licensed under the Apache-2.0 License.

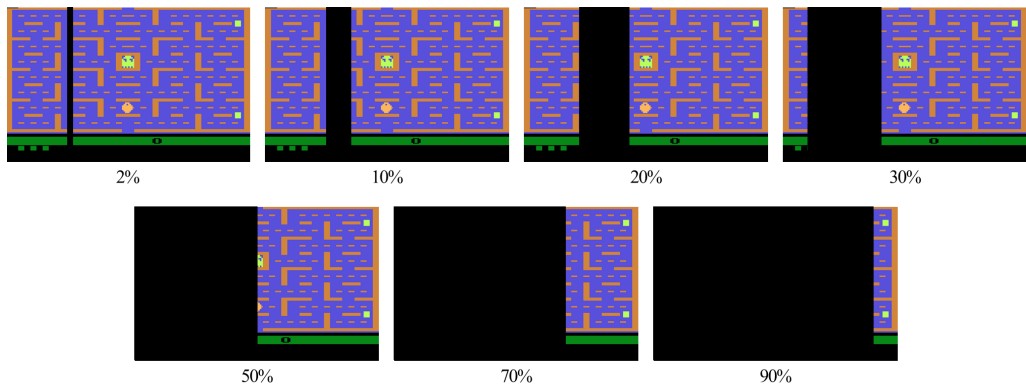

Figure 6: An example of maximally erased masking results as the scale and ratio change.

and projector, both parameterized by $\phi$. The IT network is composed of a CNN, transformer, and projector, all parameterized by $\xi$. IT and MT networks share the same architecture as the online network. The transformer (Vaswani et al., 2017) included in online and IT networks have two blocks with two self-attention heads. We used positional embedding as the input of the transformer to consider the relative position of consecutive state information output from the CNN. Both the predictor and projector are composed of a multi-layer perceptron (MLP). The DRL method for learning Atari uses a Q-learning head consisting of an MLP and three convolutional layers with rectified linear units (ReLU) activation functions, the same as the original architecture of Rainbow (Hessel et al., 2018), based on CURL and MLR. For the DMControl benchmark, the encoder (e.g., critic) consists of four convolutional layers with ReLU followed by a layer normalization (Ba et al., 2016). The Q-learning head for policy learning is MLP.

## B.2 TRAINING DETAILS

**Training** We conducted training by weighting the $\mathcal{L}_{rl}$ loss function of DRL methods and the $(\mathcal{L}_{recon} + \mathcal{L}_{inverse})$ loss function of MIND by $\lambda$. Algorithm 1 summarizes the training process of our method. Because our method performs end-to-end training, we set the batch size to be the same as that in CURL. Moreover, following the approaches of SPR (Schwarzer et al., 2021a) and MLR (Yu et al., 2022) in Atari and DMControl benchmarks, we started training when a certain number of transitions were accumulated in the replay buffer. We trained the online network and Q-learning head using the Adam optimizer (Kingma & Ba, 2014) with settings $(\beta_1, \beta_2) = (0.9, 0.999)$ and $\epsilon = 0.00015$. The learning rate is 0.0001. On DMControl applying SAC, the learning temperature parameter $\alpha$ is set to $(\beta_1, \beta_2) = (0.5, 0.999)$. The learning rate for temperature $\alpha$ is 0.0001, otherwise, it is fixed at 0.001.

**Hyperparameters** Table 8 and Table 9 present the full set of hyperparameters used in the Atari and DMControl 100k benchmarks. It is worth noting that we largely maintained the hyperparameter values used for training the Rainbow and SAC in previous studies (Laskin et al., 2020a; Schwarzer et al., 2021a; Yu et al., 2022). The main hyperparameters of MIND, such as input sequence length, transformer depth, and number of heads, were set to six, two, and two, respectively. The masking strategy was configured with a minimum value of 2% (scale of 0.02 and ratio of 0.3) and a maximum value of 20% (scale of 0.33 and ratio of 3.3). These hyperparameters were applied identically to both Atari and DMControl benchmarks. The loss weight $\lambda$ was set to one for all benchmarks, except for several Atari games. Breakout, Pong, and Up and Down, where small objects are a factor or rapidly changing states, worked well when $\lambda = 5$. The momentum coefficient $\beta$ for updating the IT and MT networks was set to 0.999 in Atari and 0.95 in DMControl.

**Random erasing** We used random erasing, provided by the open-source kornia augmentation[4] (Riba et al., 2020), as a masking strategy. This technique erases areas in rectangular shapes accord-

---

[4]Link: `https://kornia.readthedocs.io/en/latest/augmentation.html`, licensed under the Apache-2.0 License.

Table 3: Wall-clock time results of the methods trained in the Atari and DMControl environments with 100k steps on a single RTX 3090 GPU. SPR, PlayVirtual, and MLR wall-clock times used results reported in previous methods.

| Method | Runtime in Hours for Atari 100k | Runtime in Hours for DMControl 100k |
|---|---|---|
| SPR | 6.0 | 4.3 |
| PlayVirtual | 10.9 | 5.2 |
| MLR | 8.2 | 6.5 |
| MLR (6) | **3.6** | **2.8** |
| MIND | 4.4 | 3.3 |

ing to important parameters: scale and ratio. It generates data with various levels of occlusion to build a robust model (Zhong et al., 2020). The scale refers to the range of proportion of the erased area relative to the input frame, and the ratio refers to the range of aspect ratio of the erased area. Both parameters have a range, bounded by minimum and maximum values. This method is applied with uniform sampling within a range of scale and ratio values at each training iteration. We fixed the minimum values of the scale and ratio, adjusted only the maximum values, and then determined the optimal masking ratio. Figure 6 shows an example of maximal erasure as we changed scale and ratio in Ms Pacman. Based on the state of $84 \times 84 \times 4$, a 2% erasure results from scale and ratio values of 0.02 and 0.3. The 10% erasure sets the scale and ratio to 0.15 and 1.5, 20% to 0.33 and 3.3, and 30% to 0.55 and 3.5. A 50% erasure sets the scale and ratio to 0.7 and 7.0, 70% to 1.1 and 9.0, and 90% to 1.7 and 9.9. In Figure 6, 2% and 10% have less effect on factors such as agents, enemies, obstacles, and viewpoints. On the other hand, 50%, 70%, and 90% cover most of the important factors, thereby reducing the efficiency of learning. In conclusion, our method improves performance by learning representations while preserving essential information when applying scale and ratio ranges of (0.02, 0.33) and (0.3, 3.3), meaning a minimum of 2% and a maximum of 20%, on the Atari and DMControl benchmarks, respectively.

**Wall-clock time**   We report the average wall-clock time taken to train for 100k interactions on both the Atari and DMControl benchmarks. Table 3 presents the wall-clock time results of MIND and previous methods, all of which were trained using a single RTX 3090 GPU. The MIND results reflect the time taken to train with the optimal hyperparameters identified in our ablation study, whereas the times reported for other methods are based on officially reported results. As indicated in Table 3, SPR and PlayVirtual exhibited longer training times than MIND. In particular, SPR's long training time, despite being a single-task operation, is attributable to its reliance on a CNN-based transition model that processes five timesteps iteratively. In contrast, our method handles six timesteps simultaneously using a transformer. In the case of Atari, while the transition model in SPR has 477,260 (95,452 multiplied by 5) parameters, our transformer with both input and output dimensions set to 128 contains 198,400 parameters. Similar patterns were observed in the DMControl results. MLR, similar to the masked modeling of MIND, approximately doubled the runtime compared to MIND. When calculating MLR with an input sequence length of six, it took less time than MIND. However, when fine-tuning for optimal performance, our method outperformed MLR in terms of time complexity, especially considering MLR's optimal input sequence length of 16. In conclusion, these results affirm that MIND is competitive in terms of time complexity and computational efficiency.

## C   MORE EXPERIMENTAL RESULTS

### C.1   ATARI-100K RESULTS

We present the scores of all methods for 26 games on Atari 100k in Table 4. Comparative methods include DER (Van Hasselt et al., 2019), OTR (Kielak, 2019), CURL (Laskin et al., 2020a), DrQ (Yarats et al., 2021b), SPR (Schwarzer et al., 2021a), PlayVirtual (Yu et al., 2021), and MLR (Yu et al., 2022). DER, OTR, CURL, DrQ, and SPR reported average scores based on 100 random seeds. PlayVirtual, MLR, and MIND reported results based on 15, three, and ten random seeds, respectively. All methods, excluding MLR and MIND, used scores provided by rliable (Agarwal et al., 2021). Each seed was evaluated using the average score across 100 episodes. Our method

Table 4: Performance results of the human scores, previous methods, and MIND for 26 games on Atari 100k. The best values per game between methods are in bold.

| Game | Random | Human | DER | OTR | CURL | DrQ | SPR | PlayVirtual | MLR | MIND |
|---|---|---|---|---|---|---|---|---|---|---|
| Alien | 227.8 | 7,127.7 | 802.3 | 570.8 | 711.0 | 734.1 | 941.9 | 947.8 | 990.1 | **1,029.3** |
| Amidar | 5.8 | 1,719.5 | 125.9 | 77.7 | 113.7 | 94.2 | 179.7 | 165.3 | **227.7** | 174.6 |
| Assault | 222.4 | 742.0 | 561.5 | 330.9 | 500.9 | 479.5 | 565.6 | 702.3 | 643.7 | **840.0** |
| Asterix | 210.0 | 8,503.3 | 535.4 | 334.7 | 567.2 | 535.6 | **962.5** | 933.3 | 883.7 | 827.1 |
| Bank Heist | 14.2 | 753.1 | 185.5 | 55.0 | 65.3 | 153.4 | **345.5** | 245.9 | 180.3 | 143.2 |
| BattleZone | 2,360.0 | 37,187.5 | 8,977.0 | 5,139.4 | 8,997.8 | 10,563.6 | 14,834.1 | 13,260.0 | **16,080.0** | 15,125.0 |
| Boxing | 0.1 | 12.1 | -0.3 | 1.6 | 0.9 | 6.6 | 35.7 | **38.3** | 26.4 | 15.0 |
| Breakout | 1.7 | 30.5 | 9.2 | 8.1 | 2.6 | 15.4 | 19.6 | **20.6** | 16.8 | 7.0 |
| Chopper Command | 811.0 | 7,387.8 | 925.9 | 813.3 | 783.5 | 792.4 | 946.3 | 922.4 | 910.7 | **1,055.0** |
| Crazy Climber | 10,780.5 | 35,829.4 | 34,508.6 | 14,999.3 | 9,154.4 | 21,991.6 | 36,700.5 | 23,176.7 | 24,633.3 | **59,145.0** |
| Demon Attack | 152.1 | 1,971.0 | 627.6 | 681.6 | 646.5 | **1,142.4** | 517.6 | 1,131.7 | 854.6 | 1,000.4 |
| Freeway | 0.0 | 29.6 | 20.9 | 11.5 | 28.3 | 17.8 | 19.3 | 16.1 | 30.2 | **31.2** |
| Frostbite | 65.2 | 4,334.7 | 871.0 | 224.9 | 1,226.5 | 508.1 | 1,170.7 | 1,984.7 | 2,381.1 | **2,704.7** |
| Gopher | 257.6 | 2,412.5 | 467.0 | 539.4 | 400.9 | 618.0 | 660.6 | 684.3 | **822.3** | 723.6 |
| Hero | 1,027.0 | 30,826.4 | 6,226.0 | 5,956.5 | 4,987.7 | 3,722.6 | 5,858.6 | **8,597.5** | 7,919.3 | 8,139.6 |
| Jamesbond | 29.0 | 302.8 | 275.7 | 88.0 | 331.0 | 251.8 | 366.5 | 394.7 | 423.2 | **546.5** |
| Kangaroo | 52.0 | 3,035.0 | 581.7 | 348.5 | 740.2 | 974.5 | 3,617.4 | 2,384.7 | **8,516.0** | 6,885.7 |
| Krull | 1,598.0 | 2,665.5 | 3,256.9 | 3,655.9 | 3,049.2 | **4,131.4** | 3,681.6 | 3,880.7 | 3,923.1 | 3,837.8 |
| Kung Fu Master | 258.5 | 22,736.3 | 6,580.1 | 6,659.6 | 8,155.6 | 7,154.5 | 14,783.2 | 14,259.0 | 10,652.0 | **27,654.6** |
| Ms Pacman | 307.3 | 6,951.6 | 1,187.4 | 908.0 | 1,064.0 | 1,002.9 | 1,318.4 | 1,335.4 | 1,481.3 | **1,660.6** |
| Pong | -20.7 | 14.6 | -9.7 | -2.5 | -18.5 | -14.3 | -5.4 | -3.0 | **4.9** | 1.5 |
| Private Eye | 24.9 | 69,571.3 | 72.8 | 59.6 | 81.9 | 24.8 | 86.0 | 93.9 | **100.0** | **100.0** |
| Qbert | 163.9 | 13,455.0 | 1,773.5 | 552.5 | 727.0 | 934.2 | 866.3 | 3,620.1 | 3,410.4 | **3,769.4** |
| Road Runner | 11.5 | 7,845.0 | 11,843.4 | 2,606.4 | 5,006.1 | 8,724.7 | 12,213.1 | 13,429.4 | 12,049.7 | **14,386.7** |
| Seaquest | 68.4 | 42,054.7 | 304.6 | 272.9 | 315.2 | 310.5 | 558.1 | 532.9 | 628.3 | **658.0** |
| Up N Down | 533.4 | 11,693.2 | 3,075.0 | 2,331.7 | 2,646.4 | 3,619.1 | **10,859.2** | 10,225.2 | 6,675.7 | 7,825.5 |
| IQM | 0.000 | 1.000 | 0.184 | 0.113 | 0.097 | 0.236 | 0.375 | 0.385 | 0.432 | **0.500** |
| OG | 1.000 | 0.000 | 0.697 | 0.829 | 0.780 | 0.679 | 0.554 | 0.552 | 0.522 | **0.489** |

demonstrated superiority in the robust evaluation metrics IQM and OG (Agarwal et al., 2021). When evaluated based on the highest score per game between methods, our method won 13 out of 26 games. Notably, MIND significantly outperformed previous methods in Crazy Climber and Kung Fu Master, games where backgrounds, viewpoints, obstacles, and objects rapidly change.

## C.2 DMCONTROL-1M RESULTS

DMControl, a benchmark for continuous tasks, includes various difficulty levels. We applied our method to five representative medium-difficulty DMControl tasks to demonstrate its effectiveness in more challenging scenarios. Our comparison includes state-of-the-art methods in medium-difficulty DMControl environments, evaluated over 1M interactions. These methods include CURL (Laskin et al., 2020a), DrQ (Yarats et al., 2021b), DrQ-v2 (Yarats et al., 2022), SPR (Schwarzer et al., 2021a), ATC (Stooke et al., 2021), and A-LIX (Cetin et al., 2022) and use the results presented in (Zheng et al., 2023). The comparison methods reported their average scores for six different seeds, while our results are based on two seeds. Table 5 shows the scores, along with the means and medians, for the five medium-difficulty DMControl tasks across all methods. Our method demonstrated superior mean and median scores in these environments. Notably, in all environments, our method outperformed ATC and A-LIX, which were previously the best-performing methods among the compared

Table 5: Performance comparison of medium-difficulty DMControl 1M. The best score of each environment is in **bold**.

| Environment | CURL | DrQ | DrQ-v2 | SPR | ATC | A-LIX | MIND |
|---|---|---|---|---|---|---|---|
| Quadruped, run | $181 \pm 14$ | $179 \pm 18$ | $407 \pm 21$ | $448 \pm 79$ | $432 \pm 54$ | $454 \pm 42$ | $\mathbf{520 \pm 30}$ |
| Walker, run | $387 \pm 24$ | $451 \pm 73$ | $517 \pm 43$ | $560 \pm 71$ | $502 \pm 171$ | $617 \pm 12$ | $\mathbf{659 \pm 18}$ |
| Hopper, hop | $152 \pm 34$ | $192 \pm 41$ | $192 \pm 41$ | $154 \pm 10$ | $112 \pm 98$ | $225 \pm 13$ | $\mathbf{293 \pm 3}$ |
| Reacher, hard | $400 \pm 29$ | $471 \pm 45$ | $572 \pm 51$ | $711 \pm 92$ | $863 \pm 12$ | $510 \pm 16$ | $\mathbf{954 \pm 28}$ |
| Acrobot, swingup | $5 \pm 1$ | $24 \pm 8$ | $210 \pm 12$ | $198 \pm 21$ | $206 \pm 61$ | $112 \pm 23$ | $\mathbf{217 \pm 13}$ |
| Mean | 225.0 | 263.4 | 379.6 | 414.2 | 423.0 | 383.6 | **528.8** |
| Median | 181.0 | 192.0 | 407.0 | 448.0 | 432.0 | 454.0 | **520.0** |

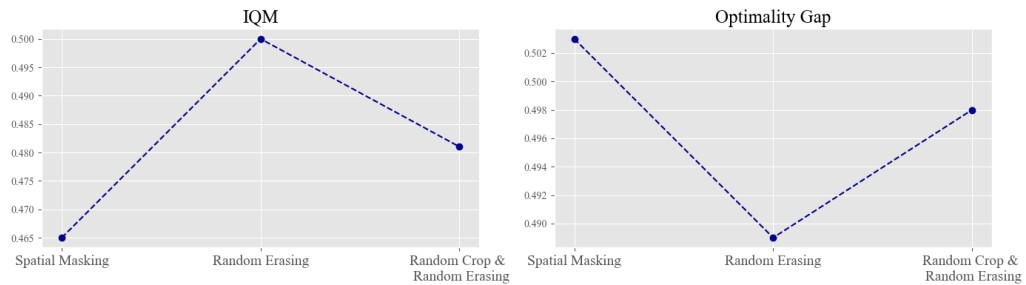

Figure 7: Comparison results of IQM and OG for different masking strategies.

algorithms. This outcome confirms our method's high data efficiency, even in more challenging settings.

### C.3 Further Analysis

**Masking strategy**  We compared spatial masking, random erasing and random cropping + random erasing. Random erasing obscures square-shaped segments of information through uniform sampling within a predetermined range of scale and ratio values at each training iteration. Figure 7 shows the IQM and OG results when the masking ratio is 20%. Our findings demonstrated that random erasing improves representation learning compared to spatial masking. Our strategy, while straightforward, is markedly effective in enhancing the data efficiency of DRL. Moreover, incorporating random cropping exhibited a decline in performance. This proves that using only random erasing without random cropping contributes to improving performance.

**Momentum coefficient**  MIND updates the target network with EMA. Figure 8 shows the aggregated results from multiple tests with different momentum coefficients (0.9, 0.99, and 0.999). We noticed that as the momentum coefficient increases, the performance notably improves.

**Impact of the target network**  MIND utilizes an asymmetric architecture based on a self-distillation approach with online and target networks. We investigated the role of the target network in inverse dynamics modeling. We tested 12 games where MIND won, as detailed in Table 4. Table 6 demonstrates the average scores from five random seeds. From our analyses, it became evident that the absence of a target network leads to rapid performance degradation because of representation collapse.

**Sequence length**  We hypothesized that an appropriate length allows us to learn useful state representations because overly short or long sequence lengths might result in trivial solutions for reconstruction and learning dynamics. We utilized various sequence lengths $T$ (2, 4, 6, 8, and 10), each accompanied by a maximum masking ratio of 20%. The transformer depth and momentum coefficient were fixed at two and 0.999. Figure 9 displays the IQM and OG results obtained by using stratified bootstrapping confidence levels from multiple tests for varying input sequence length.

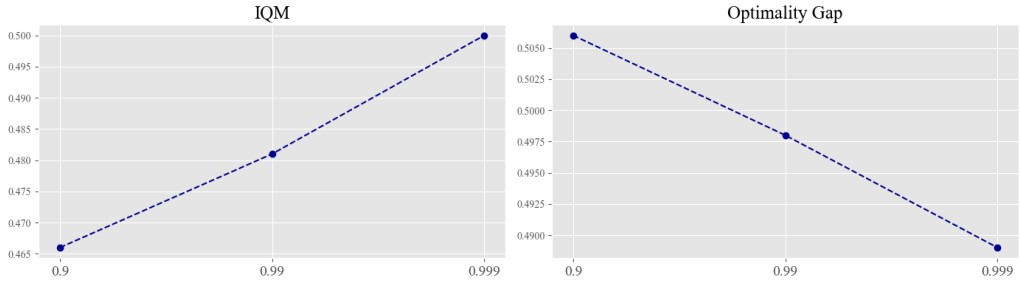

Figure 8: Comparison results of IQM and OG for different values of momentum coefficient.

Table 6: Results of the experiment on the presence or absence of a target network using only inverse dynamics modeling in MIND.

| Game | Without Target Network | With Target Network |
|---|---|---|
| Alien | 657.7 | **978.2** |
| Assault | 621.5 | **800.1** |
| Chopper Command | 466.0 | **962.0** |
| Crazy Climber | 22,796.1 | **47,522.4** |
| Freeway | 23.7 | **30.7** |
| Frostbite | 884.2 | **2,665.2** |
| Jamesbond | 224.0 | **503.0** |
| Kung Fu Master | 8,590.0 | **19,753.4** |
| MS Pacman | 830.5 | **1,292.4** |
| Qbert | 914.1 | **2,705.1** |
| Road Runner | 6,222.0 | **9,952.0** |
| Seaquest | 317.4 | **603.6** |

The experimental results indicate that increasing the input sequence length improves performance, but it diminishes when the lengths are either excessively short or long. Our method demonstrated effective performance with a sequence length of six. Therefore, in line with our hypothesis, the data efficiency of DRL improved through learning valuable state representations with an appropriate length. This length was kept to a small number, consistent with the reported SPR (Schwarzer et al., 2021a) results. In conclusion, these results also show competitiveness with previous methods considering sequence lengths in terms of computational efficiency.

**Transformer depth** Our primary goal is to learn the DRL policy from the state representation inferred by the CNN encoder. The transformer is a network independent of the CNN encoder, primarily used for reconstruction and inverse dynamics modeling tasks. Therefore, the transformer can maintain a flexible design with a lightweight architecture, as demonstrated by the design and experiments of the asymmetric encoder-decoder proposed by (He et al., 2022). We evaluated the efficiency of different transformer depths (1, 2, 4, and 6). The input sequence length, maximum masking ratio, and momentum coefficient were fixed at six, 20%, and 0.999, respectively. Figure 10 displays the aggregated results and the number of parameters from multiple tests at different transformer depths. As the transformer depth increased, all metrics decreased. This represents the data inefficiency of DRL. In conclusion, our method enhanced performance when the transformer depth was set at two, along with a relatively small number of parameters. This outcome is competitive in terms of computational cost.

## C.4 DISCUSSIONS

**Evaluation of learned representations** Our method learned representations through self-supervised multi-task learning for masked modeling and inverse dynamics modeling. Given that masked modeling and inverse dynamics modeling are regression and classification problems respec-

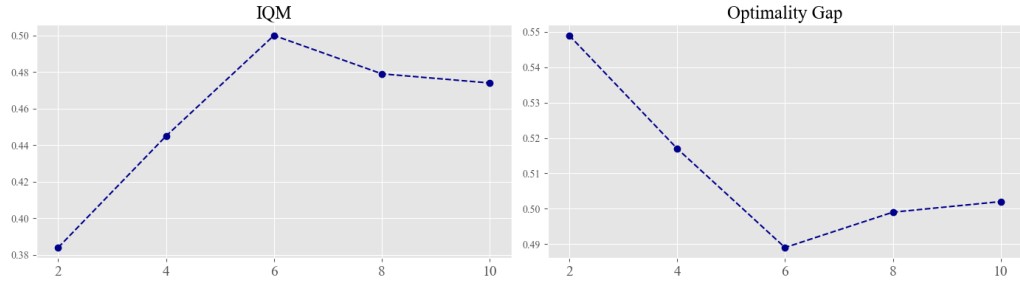

Figure 9: Comparison results of IQM and OG for different variations of input sequence length.

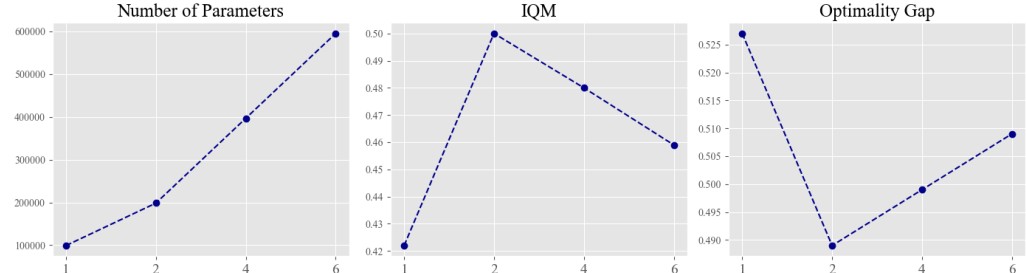

Figure 10: Comparison results of IQM, OG and number of parameters for different transformer depths of MIND.

tively, we can utilize the corresponding metrics to evaluate the learned representations. Masked modeling was verified with MSE, and inverse dynamics modeling was verified with f1-score. We validated the learned representations with Crazy Climber and Kung Fu Master, in which MIND outperformed previous methods among 26 Atari games. Table 7 shows the results of evaluating the quality of the learned representations from both modeling. The evaluations were conducted at 3k and 100k training steps. The results showed that both modelings improve the quality of the state representations as training progresses. Specifically, in Crazy Climber and Kung Fu Master, where the action space is nine and 14 respectively, it was observed that inverse dynamics modeling predicts the consecutive actions accurately. In conclusion, our method positively influenced the data efficiency of DRL by learning beneficial representations.

**The relationship between MLR and MIND** Both MLR (Yu et al., 2022) and MIND are proposed to improve the data efficiency of DRL. These methods are designed to incorporate spatio-temporal information using masking augmentation. However, while MLR bears similarities to the masking modeling included in MIND, there are also distinct differences. MLR uses both state and action information within the transformer. In contrast, MIND aims to predict consecutive states, hence actions can be considered as implicitly embedded. Therefore, we did not leverage actions as inputs according to the approach of (Lesort et al., 2018). In addition, there are differences in the masking strategy. When comparing IQM and OG, masked modeling of MIND, with scores of 0.406 and 0.535, respectively, fell short in performance compared to MLR. It is important to note that these outcomes were observed when MIND was limited to using only masked modeling, without the application of inverse dynamics. The core of our method is to effectively learn the rapidly evolving state representations of backgrounds, viewpoints, obstacles, and objects. Therefore, we addressed the limitations highlighted in MLR by complementing masked modeling with inverse dynamics modeling. When only inverse dynamics modeling was applied, the IQM and OG scores were improved to 0.461 and 0.508, respectively, surpassing the performance of MLR. This underscores the value of the inverse dynamics component in our approach.

**Why are the masking ratio results of MLR and MIND different?** While MLR employs specific spatio-temporal cubes for masking to prevent overlap in sequential states, ensuring consistent masking ratios across all states, MIND adopts a different approach. MIND utilizes a random erasing technique that indiscriminately selects a rectangular region and removes its pixels without consid-

Table 7: Evaluation results of the learned representations from masked modeling and inverse dynamics modeling.

| Method | Metric | Crazy Climber | Kung Fu Master |
|---|---|---|---|
| Masked Modeling | 3k MSE | 0.016 | 0.016 |
| | 100k MSE | **0.005** | **0.005** |
| Inverse Dynamics Modeling | 3k F1-Score | 0.053 | 0.022 |
| | 100k F1-Score | **0.840** | **0.708** |

ering the temporal aspect. This stochastic strategy is consistently applied across all training states, with masking rates varying between a minimal 2% and a maximal 20%. As a result, each state in the batch has distinct masking points. These divergent masking strategies naturally lead to different outcomes concerning the masking ratios between MLR and MIND.

**Applications and limitations**   MIND has been proposed to improve the data efficiency of DRL, and it can be utilized as an auxiliary task in other pixel-based DRL methods. Furthermore, it can be modified to learn representations or understand dynamics in fields such as computer vision, natural language processing, or other DRL research. Although our method demonstrated impressive performances on the Atari, it revealed some limitations in certain games. Because our method trains masked modeling and inverse dynamics modeling end-to-end, accurately capturing the motion of small objects, as seen in games like Breakout and Pong, can be challenging. Moreover, while our method for predicting consecutive actions in small discrete action spaces works well in games with relatively large action spaces, it falls short in learning representations for games with smaller action spaces, such as Breakout, Gopher, Pong, and Qbert. Furthermore, training the agent from scratch can lead to data inefficiency if the learning process involves incorrect reward signals and actions. Finally, the proposed method may have high real-time complexity, making it difficult to apply to real-world applications. In future work, we plan to explore and develop learning methods that can address these limitations. Potential areas for exploration include decoupling models to better capture the motion of small objects, using intrinsic motivation through dynamics modeling, or using learning methods that incorporate expert demonstrations. By refining our approach, we aim to further improve the performance and applicability of MIND across a broader range of gaming environments.

Table 8: A full set of hyperparameters used for Atari.

| Hyperparameter | Value |
|---|---|
| Image size | (84, 84) |
| Max frames per episode | 108,000 |
| Stacked frames | 4 |
| Action repeat | 4 |
| Support of Q-distribution | 51 |
| Q network: channels | 32, 64, 64 |
| Q network: filter size | $8{\times}8, 4{\times}4, 3{\times}3$ |
| Q network: hidden units | 256 |
| Non-linearity | ReLU |
| Target network: update period | 1 |
| Exploration | Noisy nets |
| Update | Distributional Q |
| Noisy nets parameter | 0.5 |
| Replay buffer size | 100,000 |
| Training steps | 100,000 |
| Minimum replay size for sampling | 2,000 |
| Reward clipping | [-1, 1] |
| Multi-step return | 10 |
| Discount factor | 0.99 |
| Batch size | 32 |
| Optimizer | Adam |
| Optimizer: learning rate | 0.0001 |
| Optimizer $\epsilon$ | 0.00015 |
| Replay period every | 1 |
| Max gradient norm | 10 |
| Priority exponent | 0.5 |
| Priority correction | 1 |
| Projection size | 128 |
| Sequence length | 6 |
| Transformer depth | 2 |
| Number of heads | 2 |
| Momentum coefficient | 0.999 |
| Mask scale | (0.02, 0.33) |
| Mask ratio | (0.3, 3.3) |

Table 9: A full set of hyperparameters used for DMControl.

| Hyperparameter | Value |
|---|---|
| Image size | (84, 84) |
| Stacked frames | 3 |
| Action repeat | 2 finger, spin and walker, walk; |
| | 8 cartpole, swingup; |
| | 4 otherwise |
| Non-linearity | ReLU |
| Initial steps | 1,000 |
| Evaluation episodes | 10 |
| Replay buffer size | 100,000 |
| Discount factor | 0.99 |
| Batch size for policy learning | 512 |
| Batch size for MIND | 128 |
| Optimizer | Adam |
| $(\beta_1, \beta_2) \rightarrow (\theta, \psi)$ | (0.9, 0.999) |
| $(\beta_1, \beta_2) \rightarrow (\alpha)$ | (0.5, 0.999) |
| Learning rate $(\theta, \psi)$ | 0.0002 cheetah, run; |
| | 0.001 otherwise |
| Learning rate $(\alpha)$ | 0.0001 |
| Initial temperature | 0.1 |
| Q-function EMA $(\tau)$ | 0.99 |
| Critic target update freq | 2 |
| Target network update period | 1 |
| State representation dimension | 50 |
| Sequence length | 6 |
| Transformer depth | 2 |
| Number of heads | 2 |
| Momentum coefficient | 0.9 walker, walk; |
| | 0.95 otherwise |
| Mask scale | (0.02, 0.33) |
| Mask ratio | (0.3, 3.3) |

