# OpenReview forum: "MIND: Masked and Inverse Dynamics Modeling for Data-Efficient Deep Reinforcement Learning"
_ICLR.cc/2024/Conference — Submitted to ICLR 2024_

### Official Review · Reviewer_Hy4i · 2023-10-31

**Soundness:** 1 poor
**Presentation:** 1 poor
**Contribution:** 2 fair
**Rating:** 3
**Confidence:** 5

**Summary:**

This work proposes to use self-supervised tasks including reconstruction and next-state prediction based on masked input states, to improve deep reinforcement learning algorithm. The proposed method also combines Transformer to jointly model the short state sequence input. The proposed method is compared to multiple baselines and exhibits better performance in Atari and DMC benchmarks.

**Strengths:**

- The proposed method is experimentally shown to be effective on two benchmarks.

- Analysis on masking strategy is extensive and informative.

**Weaknesses:**

1. **Confusing diagram**. Figure 1 is very misleading to explain "reconstruction" and "inverse dynamics modeling". The reconstruction usually refers to predicting the original image, while the actual method is about minimizing error between the latent representations from two networks. This diagram also does not help explain action prediction (inverse dynamics modeling). The current diagram is showing a comparison of image embeddings.

2. **Clarity**. Seciton 3 does not explain:

   * Why a Transformer module is introduced to help modeling
   * How action is predicted from the model, using current and next states
   * How the online network performs state representation regression and action prediction simultaneously. Are there seperate heads for each auxiliary task?

3. **Baselines**. Clearly there are missing baselines like RAD [1], DrQv2 [2] according to this study [3] on self-supervised learning with RL, and possibly [4] which uses Transformer for next frame prediciton, and vanilla MAE [5]. With the current baselines I can not evaluate the significance of proposed method, considering the novelty of the proposed method is limited.

4. Due to extra architecture and parameter introduced in the Transformer module, are the baseline models using the same/similar architecture to ensure the fairness? Is the Transformer important in the modeling?

[1] Laskin, Michael et al. “Reinforcement Learning with Augmented Data.” ArXiv abs/2004.14990 (2020)

[2] Yarats, Denis, et al. "Mastering visual continuous control: Improved data-augmented reinforcement learning." arXiv preprint arXiv:2107.09645 (2021).

[3] Li, Xiang et al. “Does Self-supervised Learning Really Improve Reinforcement Learning from Pixels?” ArXiv abs/2206.05266 (2022)

[4] Micheli, Vincent, Eloi Alonso, and François Fleuret. "Transformers are sample efficient world models." arXiv preprint arXiv:2209.00588 (2022).

[5] He, Kaiming et al. “Masked Autoencoders Are Scalable Vision Learners.” 2022 IEEE/CVF Conference on Computer Vision and Pattern Recognition (CVPR) (2021): 15979-15988.

**Questions:**

Please see weaknesses.

---

> ### Author Response · Authors · 2023-11-16
> **Response to Reviewer Hy4i**
>
> We would like to thank you for your thorough evaluation and in-depth comments. We considered all of your comments and suggestions during the rebuttal of our manuscript. All of the changes made in the revised manuscript have been highlighted in blue.
>
> **Q1: Confusing diagram.**
>
> A1: Thank you for your feedback. Our method is self-supervised multi-task learning that combines masked modeling and inverse dynamics modeling. In the context of RL, our approach, like the one in [1], is centered around reconstructing masked or missing information in the latent space, not at the pixel level of predicting the original image. This method has been shown to be effective [1]. Regrading inverse dynamics modeling, our method predicts actions using a classifier based on latent representations from online and IT networks. Acknowledging the reviewer's concern, we agree that Figure 1 might not clearly represent this aspect. We have revised Figure 1 to better illustrate both masked and inverse dynamics modeling, making it easier for readers to understand our method.
>
> [1] Yu, T., Zhang, Z., Lan, C., Lu, Y., & Chen, Z. (2022). Mask-based latent reconstruction for reinforcement learning. Advances in Neural Information Processing Systems, 35, 25117-25131.
>
> **Q2: Clarity.**
>
> A2: To address the high correlation among consecutive frames in RL, we introduced a transformer. It helps capture spatio-temporal information effectively while considering high correlation.
>
> For inverse dynamics modeling, actions are predicted using a classifier. This classifier utilizes the current state representations from the online network and the next state representations from the IT network. We have revised the manuscript to clearly mentioned in the manuscript, enhancing clarity on how actions are predicted.
>
> Regrading the online network, it simultaneously performs state representation regression for the reconstruction task and action prediction for inverse dynamics modeling. The state representations output from the online and IT networks are used to predict actions through a classifier, while the online and MT networks focus on minimizing state representation errors. We have separate neural networks for each task, ensuring clear task-specific processing.
>
> **Q3: Baselines.**
>
> A3: In response to the reviewer's suggestion, we have included comparison in our revised Table 2. We conducted experiments across six DMControl environments with a limit of 100k interactions. Our method demonstrated notable performance, winning in three out of six environments and outperforming both the mean and median across all environments. Furthermore, we experimented on five tasks of medium-difficulty DMControl with 1M interactions, choosing comparison methods in line with the model-free approach. The comparison methods reported their average scores for six different seeds, while our results are based on two seeds. The results show improved data efficiency of our method at medium-difficulty compared to the baselines. We will conducted additional experiments to ensure a fair comparison and report the results in the final version. Detailed results are described in Appendix C.2.
>
> **Q4: Due to extra architecture and parameter introduced in the transformer module, are the baseline models using the same/similar architecture to ensure the fairness? Is the transformer important in the modeling?**
>
> A4: Our baseline models include transformer-based and transformer-free methods. Among the baseline models, MLR [1] used a transformer with a similar structure to our method. The use of a transformer is crucial in our method due to the high correlation among consecutive frames in RL. This correlation needs a mechanism that can effectively capture spatio-temporal information. Transformers, and other sequence processing architecture like recurrent methods, are important for handling these correlations. In comparison, methods such as CURL [2] and DrQ [3], which do not adequately address this high correlation, tend to be less data-efficient. Therefore, the inclusion of a transformer in our model is not only important for fair comparison but also essential for the effectiveness of our approach.
>
> [1] Yu, T., Zhang, Z., Lan, C., Lu, Y., & Chen, Z. (2022). Mask-based latent reconstruction for reinforcement learning. Advances in Neural Information Processing Systems, 35, 25117-25131.
>
> [2] Laskin, M., Srinivas, A., & Abbeel, P. (2020, November). Curl: Contrastive unsupervised representations for reinforcement learning. In International Conference on Machine Learning (pp. 5639-5650). PMLR.
>
> [3] Yarats, D., Kostrikov, I., & Fergus, R. (2020, October). Image augmentation is all you need: Regularizing deep reinforcement learning from pixels. In International conference on learning representations.

---

### Official Review · Reviewer_t6qc · 2023-10-31

**Soundness:** 2 fair
**Presentation:** 3 good
**Contribution:** 3 good
**Rating:** 5
**Confidence:** 4

**Summary:**

This paper introduces a novel representation learning objective MIND that integrates inverse dynamics modeling with the masked modeling of consecutive states. This dual approach enables the representation to encapsulate both the agent-controllable aspects as well as static features within the image. The effectiveness of this method is demonstrated through empirical results on the Atari 100K and DMControl-100K benchmarks.

**Strengths:**

The paper is very well-written. The empirical evaluation is comprehensive for tasks with discrete action spaces Atari-100K. Additionally, comprehensive ablation studies of the learning objective as well as hyperparameter choices are conducted.

**Weaknesses:**

The evaluation for continuous control tasks are very limited. DMControl 100K is originally proposed and evaluated in CURL, which consists of the simplest tasks in DMControl. Later works such as Dreamer-v2/v3, DrQ-v2, A-LIX[1], ATC[2], and TACO[3] mainly consider medium-difficulty tasks as well as the more challenging humanoid domain. Thus, more tasks and especially harder medium-difficulty tasks should be evaluated for the proposed method. I recommend the author to provide a comparison at 1M/2M of harder tasks instead of the six tasks presented in the paper. Also, A-LIX[1], ATC[2], and TACO[3] should be discussed and compared in the empirical evaluation.

**Questions:**

As shown by prior works such as [4, 5], the single-step inverse model is theoretically and empirically non-sufficient to capture the full agent-centric representation. For example, in an empty gridworld, a pair of positions separated by two or more spaces may be assigned an identical representation without incurring additional loss in a one-step inverse model.. In contrast, a multi-step inverse model is both theoretically sufficient and in practice achieves good performance. Would MIND benefit from multi-step inverse modeling instead of single-step inverse?

I am willing to raise the score if my above two questions/concerns (a more comprehensive evaluation of continuous control tasks and multi-step inverse model instead of single-step) are addressed. I understand that given the time constraint of rebuttal, the authors are not able to address them fully. But it would be great at least to see the additional experiments on a few medium DMControl tasks.

**Additional References that are not included in the paper**
- [1] Edoardo Cetin, Philip J. Ball, Steve Roberts, Oya Celiktutan, Stabilizing Off-Policy Deep Reinforcement Learning from Pixels, ICML 2022
- [2] Adam Stooke, Kimin Lee, Pieter Abbeel, Michael Laskin, Decoupling Representation Learning from Reinforcement Learning, ICML 2021
- [3] Ruijie Zheng, Xiyao Wang, Yanchao Sun, Shuang Ma, Jieyu Zhao, Huazhe Xu, Hal Daumé III, Furong Huang. TACO: Temporal Latent Action-Driven Contrastive Loss for Visual Reinforcement Learning, NeurIPS 2023
- [4] Riashat Islam, Manan Tomar, Alex Lamb, Yonathan Efroni, Hongyu Zang, Aniket Didolkar, Dipendra Misra, Xin Li, Harm van Seijen, Remi Tachet des Combes, John Langford, Principled Offline RL in the Presence of Rich Exogenous Information. ICML 2023
- [5] Alex Lamb, Riashat Islam, Yonathan Efroni, Aniket Didolkar, Dipendra Misra, Dylan Foster, Lekan Molu, Rajan Chari, Akshay Krishnamurthy, John Langford, Guaranteed Discovery of Control-Endogenous Latent States with Multi-Step Inverse Models.

---

> ### Author Response · Authors · 2023-11-16
> **Response to Reviewer t6qc**
>
> We would like to thank you for your thorough evaluation and helpful suggestions and comments. We address your concerns in detail below. All of the changes made in the revised manuscript have been highlighted in blue.
>
> **Q1: The evaluation for continuous control tasks are very limited.**
>
> A1: Thank you for the valuable suggestions. We agree with the reviewer that further experiments with continuous control benchmarks is necessary. TACO believes that there is no obligation to compare because it is considered a recent work according to the policy of the conference (https://iclr.cc/Conferences/2024/ReviewerGuide). Nevertheless, to address the reviewer's concerns, we tested our method on five medium-difficulty DMControl tasks using 1M interactions, as reported by TACO. We selected these tasks to align with our model-free approach and make a comparable analysis with comparison methods. The comparison methods reported their average scores for six different seeds, while our results are based on two seeds. As shown in the table below, the results show improved data efficiency of our method at medium-difficulty compared to the benchmarks. In particular, in the environments Walker, run; Hopper, hop and Reacher, hard, our method outperformed TACO, which was the best performing method among the compared algorithms. We will conduct additional experiments to ensure a fair comparison and report the results in the final version. We have included the detailed DMControl-1M results excluding TACO in Appendix C.2 for further reference.
>
> | Environment | CURL | DrQ | DrQ-v2 | SPR | ATC | A-LIX | TACO | MIND |
> | :--- | :---: | :---: | :---: | :---: | :---: | :---: | :---: | :---: |
> | Quadruped, run | 181 $\pm$ 14 | 179 $\pm$ 18 | 407 $\pm$ 21 | 448 $\pm$ 79 | 432 $\pm$ 54 | 454 $\pm$ 42 | **541 $\pm$ 38** | 520 $\pm$ 30 |
> | Walker, run | 387 $\pm$ 24 | 451 $\pm$ 73 | 517 $\pm$ 43 | 560 $\pm$ 71 | 502 $\pm$ 171 | 617 $\pm$ 12 | 637 $\pm$ 21 | **659 $\pm$ 18** |
> | Hopper, hop | 152 $\pm$ 34 | 192 $\pm$ 41 | 192 $\pm$ 41 | 154 $\pm$ 10 | 112 $\pm$ 98 | 225 $\pm$ 13 | 261 $\pm$ 52 | **293 $\pm$ 3** |
> | Reacher, hard | 400 $\pm$ 29 | 471 $\pm$ 45 | 572 $\pm$ 51 | 711 $\pm$ 92 | 863 $\pm$ 12 | 510 $\pm$ 16 | 883 $\pm$ 63 | **954 $\pm$ 28** |
> | Acrobot, swingup | 5 $\pm$ 1 | 24 $\pm$ 8 | 210 $\pm$ 12 | 198 $\pm$ 21 | 206 $\pm$ 61 | 112 $\pm$ 23 | **241 $\pm$ 21** | 217 $\pm$ 13 |
> | Mean | 225.0 | 263.4 | 379.6 | 414.2 | 423.0 | 383.6 | 512.6 | **528.8** |
> | Median | 181.0 | 192.0 | 407.0 | 448.0 | 432.0 | 454.0 | **541.0** | 520.0 |
>
> **Q2: Would MIND benefit from multi-step inverse modeling instead of single-step inverse?**
>
> A2: Previous studies have demonstrated the effectiveness in capturing comprehensive agent-centric representations. Our method, while not direct multi-step approach, integrates elements of this concept. We input states from time *t* to *t+n* into the online network, and the next states from time *t+1* to *t+n+1* into the target network. This setup allows us to predict consecutive actions from time *t* to time *t+n*. In addition, the transformers were used in each network to process highly correlated consecutive frames in RL. A transformer can efficiently summarize information across the input sequence through its attention mechanism. Therefore, while our method differs from traditional multi-step modeling, it effectively captures past or future state dependencies, aligning with the principles of multi-step inverse dynamics.
>
> **Q3: Additional references.**
>
> A3: We have reviewed them and found their insights valuable for our work. We have updated our literature review to include the key contributions.

---

> > ### Comment · Reviewer_t6qc · 2023-11-23
> >
> > Regarding Q1, it is probably okay if you don't compare with them. But if you directly use the results from the TACO paper, you should clearly cite and mention it in your paper.
> >
> > Regarding Q2, I don't agree the objective used in the paper is doing multistep inverse because for example, to predict action $a_t$, one could only use the information of $s_t$ and $s_{t+1}$ which are coming from the frame stack. But for multistep inverse, you are only given $s_t$ and $s_{t+K}$, and no information about the intermediate states are provided.
> >
> > I will keep my score.

---

> > > ### Author Response · Authors · 2023-11-23
> > > **Response to Reviewer t6qc**
> > >
> > > Thank you for your comments. We appreciate your efforts in reviewing our work.
> > >
> > > We agree that we should cite the TACO paper because we use the results as stated by the reviewer. Therefore, we have referenced the TACO in Appendix C.2 of the revised manuscript.
> > >
> > > Regarding Q2, the attention mechanism of the transformer has the effect of making predictions using even distant information. Our method is a self-supervised multi-task learning framework that introduces a transformer to reflect spatio-temporal information while considering the high correlation of consecutive states of RL, and simultaneously learns masked reconstruction and inverse dynamics modeling. Therefore, while we appreciate the reviewer's suggestions, we believe that multi-step inverse modeling is currently out of scope for our method.

---

### Official Review · Reviewer_RYme · 2023-10-31

**Soundness:** 3 good
**Presentation:** 2 fair
**Contribution:** 2 fair
**Rating:** 6
**Confidence:** 4

**Summary:**

This paper proposes a hybrid objective of something very similar to BYOL (predicting representations of a target network) while the online network is masked, while combining this with a one-step inverse model.  This approach has nice empirical results and solid empirical analysis on Atari 100k.  The big downside I see to this paper is that the method is not extremely novel and the justifications are ad-hoc, whereas the field of representations for RL has seen rapid progress in theoretical analysis, so it should be possible to provide more detailed justifications for the technique.  Nonetheless, I see this paper as a solid empirical contribution.

**Strengths:**

-The improvements over reasonable baselines like SPR and DrQ are considerable on the Atari 100k, which is a fairly difficult setting.  The improvements on DM control suite are also convincing.
  -The analysis of what the model is learning is reasonably thorough.
  -I appreciated the study of wall-clock time, showing that the method is practical to use on a single GPU and is cheaper than other methods.

**Weaknesses:**

-The justification in this paper for using inverse dynamics along with masked prediction is fairly ad-hoc and informal.  This isn't the end of the world, but the understanding of this area from theoretical RL is rapidly advancing (Efroni 2022, Lamb 2022, for example).  There have also been a few empirical papers with related ideas such as the InfoPower paper as well as the ACRO paper (Islam 2022).  One simple thing that might be worth trying is for the IT model, use the observation k steps in the future (with k sampled U(1,5) for example) and then predict the first action.  Some theoretical work has suggested the value of this approach.
  -The basic approach seems similar to combining BYOL (with masking as the augmentation) with a one step inverse model.

**Questions:**

-Why is there an IT network, rather than reusing the MT network for the IT network, perhaps with a distinct head for predicting inverse dynamics?
  -Why not also use masking of the inputs to the IT network?

---

> ### Author Response · Authors · 2023-11-16
> **Response to Reviewer RYme**
>
> Thank you for your positive and in-depth comments about our research. We address your concerns in detail below. All of the changes made in the revised manuscript have been highlighted in blue.
>
> **Q1: The justification in this paper for using inverse dynamics along with masked prediction is fairly ad-hoc and informal.**
>
> A1: Thank you for your meaningful comments. The references that the reviewer mentioned have proven the effectiveness of multi-step dynamics modeling. Our method, while not directly adopting the multi-step method, captures similar concepts. The inverse dynamics modeling we propose is designed from the perspective of sequence modeling of the dynamics between consecutive states and actions. We used states from time *t* to *t+n* in the online network and the next states from time *t+1* to *t+n+1* in the target network. Our design predicts the consecutive actions from time *t* to time *t+n*. Because of the high correlation among consecutive frames in RL, we incorporated transformers in each network. These transformers use their attention mechanism to integrate information across the input sequence states. This step, although not a straightforward multi-step approach, effectively captures past or future state dependencies in terms of representation learning. The transformer is a sequence module that has proven to be excellent not only in the field of natural language processing but also in RL. We have enriched subsection 2.3 in our manuscript to include these relevant references and further clarify our method.
>
> **Q2: The basic approach seems similar to combining BYOL (with masking as the augmentation) with a one-step inverse model.**
>
> A2: Our method shares similarities with BYOL in terms of the learning strategy. However, to better handle the unique challenges in RL, we adopted a transformer. This addition is crucial for capturing dependencies on past or future states. It is a technique similar to multi-step inverse dynamics modeling, tailored for learning rich representations of consecutive frames.
>
> **Q3: Why is there an IT network, rather than reusing the MT network for the IT network, perhaps with a distinct head for predicting inverse dynamics?**
>
> A3: The MT and IT networks serve distinct purposes in our method. The MT network is designed for mask-based reconstruction, aiming to reconstruct missing state information at a specific point in time. On the other hand, the IT network focuses on inverse dynamics modeling. It outputs representations of the next states that is used to predict actions. This division of tasks is essential for our self-supervised multi-task learning approach, where each network is specialized for its special functionality.
>
> **Q4: Why not also use masking of the inputs to the IT network?**
>
> A4: Our method uses masking augmentation exclusively in the online network for the reconstruction task. This is to focus on missing information without the need for separate data augmentation for learning visual representations. On the other hand, the IT network is tailored for inverse dynamics modeling. It extracts representations for a series of next states, capturing essential information about rapidly changing states. While our method conducts both reconstruction and inverse dynamics modeling simultaneously, the objectives differ. Therefore, we input the original or unmasked data into the IT network to align with its specific purpose.

---

### Official Review · Reviewer_uqBc · 2023-11-01

**Soundness:** 2 fair
**Presentation:** 3 good
**Contribution:** 2 fair
**Rating:** 5
**Confidence:** 4

**Summary:**

This paper proposes to learn representations for reinforcement learning  using a combination of inverse dynamics training and masked modeling.

**Strengths:**

- The problem of learning representations from reinforcement learning makes sense
- The paper appears to improve performance over baselines
- The paper evaluates across a set of different environments, showing improved performance across them
- The paper runs a comprehensive set of ablations

**Weaknesses:**

- In equation one, p, c, q,  is not defined
- It would be good to report confidence intervals in results, for instance in Table 1
- The paper doesn't seem very novel -- it uses momentum contrast masked loss + a standard inverse dynamics representation learning objective

**Questions:**

Can the authors provide some intuition why masking + momentum contrast outperforms other baseline methods?

---

> ### Author Response · Authors · 2023-11-16
> **Response to Reviewer uqBc**
>
> Thank you for your meaningful comments on our method and experiment. We considered all of your comments and suggestions during the revision of our manuscript. All of the changes made in the revised manuscript have been highlighted in blue.
>
> **Q1: In equation one, *c*, *t*, *p*, and *q* are not defined.**
>
> A1: Thank you for your careful reading. In equation one, the CNN, transformer, projector, and predictor are denoted by *c*, *t*, *p*, and *q*, respectively. We have provided description of these terms in Section 3.
>
> **Q2: It would be good to report confidence intervals in results, for instance in Table 1.**
>
> A2: Thank you for your suggestion. We have included the standard deviations of the human-normalized scores for each game in Table 1. In addition, Figures 2 and 3, as well as Table 2, provide information about the confidence interval or standard deviation.
>
> **Q3: The paper doesn't seem very novel.**
>
> A3: Mask-based reconstruction and dynamics modeling are established techniques in representation learning. However, their application in RL, especially in rapidly changing environments, is still largely unexplored. Our work is novel in that we are among the first to combine dynamics modeling with masked modeling in a self-supervised multi-task learning framework. This approach specifically addresses the limitations noted in previous methods (referenced in [1]). Our focus extends beyond just learning visual representations. We also aim to capture the controllable elements in environments with rapid state changes. To handle the high correlation between consecutive frames in RL, we introduced a transformer. This aids in predicting a series of actions by analyzing correlations and spatio-temporal information of consecutive states. Furthermore, we conducted systematic experiments to establish best practices in this area. Our findings show that integrating inverse dynamics modeling in latent space, like MLR [1], enhances efficiency in rapidly evolving environments. This advancement in methodology represents a significant contribution to the field.
>
> [1] Yu, T., Zhang, Z., Lan, C., Lu, Y., & Chen, Z. (2022). Mask-based latent reconstruction for reinforcement learning. Advances in Neural Information Processing Systems, 35, 25117-25131.
>
> **Q4: Can the authors provide some intuition why masking + momentum contrast outperforms other baseline methods?**
>
> A4: Data augmentation techniques work very sensitively in RL. In previous studies [1][2], they empirically experimented with various data augmentations and reported which augmentation can deliver the correct reward signal to the agent. On the other hand, masked modeling, which only considers masking augmentation, is a simple yet effective way to reconstruct missing data. Typical reconstruction tasks make predictions at the pixel-level. However, in RL that leverages latent representations to learn the agent's policy, it is important to reconstruct missing state information in the latent space. In terms of learning representations for states, momentum contrast (i.e., self-distillation approach [5]) has been shown to excellent at preventing representation collapse, benefiting from its asymmetric architecture and exponential moving average updating [3][4][5]. SPR [3], demonstrated that without a target network, performance declines due to representation collapse. MLR [6] combines the strengths of both masking and momentum contrast to reconstruct latent representations, proving to be more effective than previous studies.
>
> [1] Yarats, D., Kostrikov, I., & Fergus, R. (2020, October). Image augmentation is all you need: Regularizing deep reinforcement learning from pixels. In International conference on learning representations.
>
> [2] Laskin, M., Lee, K., Stooke, A., Pinto, L., Abbeel, P., & Srinivas, A. (2020). Reinforcement learning with augmented data. Advances in neural information processing systems, 33, 19884-19895.
>
> [3] Schwarzer, M., Anand, A., Goel, R., Hjelm, R. D., Courville, A., & Bachman, P. (2020, October). Data-Efficient Reinforcement Learning with Self-Predictive Representations. In International Conference on Learning Representations.
>
> [4] Grill, J. B., Strub, F., Altché, F., Tallec, C., Richemond, P., Buchatskaya, E., ... & Valko, M. (2020). Bootstrap your own latent-a new approach to self-supervised learning. Advances in neural information processing systems, 33, 21271-21284.
>
> [5] Balestriero, R., Ibrahim, M., Sobal, V., Morcos, A., Shekhar, S., Goldstein, T., ... & Goldblum, M. (2023). A cookbook of self-supervised learning. arXiv preprint arXiv:2304.12210.
>
> [6] Yu, T., Zhang, Z., Lan, C., Lu, Y., & Chen, Z. (2022). Mask-based latent reconstruction for reinforcement learning. Advances in Neural Information Processing Systems, 35, 25117-25131.

---

> ### Comment · Reviewer_uqBc · 2023-11-28
> **Rebuttal Response**
>
> Thank you for your rebuttal response -- it helps clarify the concerns I had earlier. However, given the concerns expressed by other reviewers, I would like to maintain my score.

---

### Comment · Area_Chair_Vcjk · 2023-11-23
**Author-Reviewer discussion period ending *very* soon**

Thank you to the reviewer t6qc for responding. The authors have put great effort into their response, so can I please urge the other reviewers to answer the rebuttal.
Thank you!

---

### Meta-Review · Area_Chair_Vcjk · 2023-12-06

**Metareview:**

The paper proposes a method called masked and inverse dynamics modeling (MIND) that addresses the challenge of learning evolving state representations in pixel-based deep reinforcement learning. Reviewer uqBc highlights the paper's improvement over baselines across various environments and the comprehensive ablations but notes some missing definitions and suggests reporting confidence intervals in results. Reviewer RYme appreciates the method's considerable improvements on Atari and DM control suite benchmarks, as well as the analysis of the model's learning process and its practicality in terms of computational resources. However, concerns arise about the informal justification of combining inverse dynamics and masked prediction and the similarity to existing approaches. Reviewer t6qc praises the paper's thoroughness in empirical evaluation and ablation studies but suggests extending the evaluation to include harder continuous control tasks and comparing with other relevant methods. Reviewer Hy4i acknowledges the method's effectiveness on benchmarks but raises concerns about the clarity of explanations, and claimed that the contribution of this work is marginal --- stating that the modifications like Transformer, auxiliary tasks, and state-only embeddings are not well ablated.

In summary, while the paper showcases improvements with MIND and conducts thorough empirical evaluations, it faces challenges regarding clarity, comparisons with existing methods, and concerns about certain introduced components like the Transformer module. Following the rebuttal, there wasn't a clear consensus among reviewers for acceptance, with one advocating for rejection based on perceived marginal contributions and inadequately ablated modifications.

**Justification For Why Not Higher Score:**

Following the rebuttal, there was no clear push by reviewers for the paper to be accepted, with one reviewer actively pushing for rejection.

**Justification For Why Not Lower Score:**

N/A

---

### Decision · Program_Chairs · 2024-01-16

Reject